# SurFhead: Affine Rig Blending for Geometrically Accurate 2D Gaussian Surfel Head Avatars

**Jaeseong Lee**[1][*]  **Taewoong Kang**[1][*]  **Marcel C. Bühler**[2]  **Min-Jung Kim**[1]
**Sungwon Hwang**[1]  **Junha Hyung**[1]  **Hyojin Jang**[1]  **Jaegul Choo**[1]
[1]KAIST  [2]ETH Zürich

https://summertight.github.io/SurFhead

## Abstract

Recent advancements in head avatar rendering using Gaussian primitives have achieved significantly high-fidelity results. Although precise head geometry is crucial for applications like mesh reconstruction and relighting, current methods struggle to capture intricate geometric details and render unseen poses due to their reliance on similarity transformations, which cannot handle stretch and shear transforms essential for detailed deformations of geometry. To address this, we propose **SurFhead**, a novel method that reconstructs riggable head geometry from RGB videos using 2D Gaussian surfels, which offer well-defined geometric properties, such as precise depth from fixed ray intersections and normals derived from their surface orientation, making them advantageous over 3D counterparts. SurFhead ensures high-fidelity rendering of both normals and images, even in extreme poses, by leveraging classical mesh-based deformation transfer and affine transformation interpolation. SurFhead introduces precise geometric deformation and blends surfels through polar decomposition of transformations, including those affecting normals. Our key contribution lies in bridging classical graphics techniques, such as mesh-based deformation, with modern Gaussian primitives, achieving state-of-the-art geometry reconstruction and rendering quality. Unlike previous avatar rendering approaches, SurFhead enables efficient reconstruction driven by Gaussian primitives while preserving high-fidelity geometry.

## 1 Introduction

The construction of personalized head avatars has seen rapid advancements in both research and industry. Among the most notable developments in this field is the Codec Avatar family (Ma et al., 2021; Saito et al., 2024), which aims to reconstruct highly detailed, movie-quality head avatars using high-cost data captured from head-mounted cameras or studios. This approach has spurred significant research efforts to bridge the gap between high-cost and low-cost capture systems by utilizing only using RGB video setups. Neural Radiance Fields (NeRFs) (Mildenhall et al., 2021) have further accelerated these efforts with their topology-agnostic representations. As a result, numerous NeRF-based methods (Gafni et al., 2021; Athar et al., 2022; Zielonka et al., 2023b) for constructing head avatars from RGB videos have emerged, demonstrating potentials of improving high-cost systems (Ma et al., 2021; Yang et al., 2023; Saito et al., 2024).

Recently, 3D Gaussian Splatting (3DGS) (Kerbl et al., 2023) has been used to create photo-realistic head avatars. However, there has been no attempt to create geometrically accurate head avatars within the 3DGS framework. While other representation-based methods (Bharadwaj et al., 2023; Zheng et al., 2022; 2023; Grassal et al., 2022) demonstrate plausible geometry using implicit or explicit representations, they still suffer from suboptimal results; over-smoothed geometry from implicit neural representations or inherently limited explicit representations such as 3D Morphable Face Model (3DMFM) learned meshes or inflexible points.

In response, we introduce **SurFhead**, the first geometrically accurate head avatar model within the Gaussian Splatting framework (Kerbl et al., 2023), designed to capture deformation of head ge-

---

[*]Equal contribution

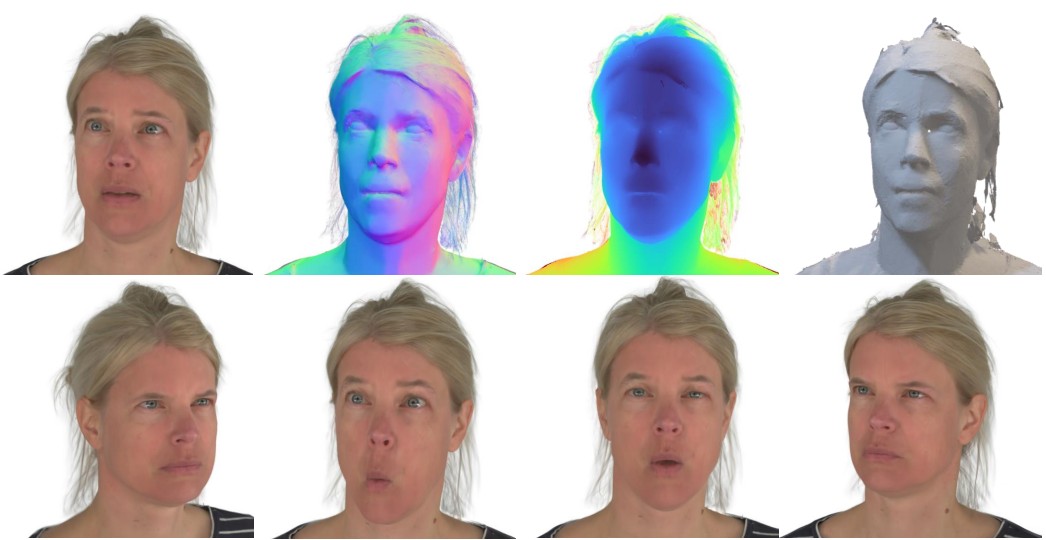

Figure 1: **SurFhead** reconstructs photo-realistic head avatars and high-fidelity surface normals, depth, and meshes from RGB videos alone. These avatars are represented through affine rigging of 2D surfel splats bound to a parametric morphable face model. **SurFhead** can fully control poses, expressions, and viewpoints, enhancing both appearance and geometry.

ometry. Our method integrates intricate affine rigging by combining Gaussians and their normals using only RGB videos. Building on 2DGS (Huang et al., 2024) with 2D surfel disks and adopting 3DMFM mesh binding from prior works (Qian et al., 2024; Shao et al., 2024; Zielonka et al., 2023b; Lombardi et al., 2021), we address limitations of previous Gaussian-based methods that rely solely on rigid, isotropic transformations. This strategy often leads to undesirable deformations during pose extrapolation when 3DMFM meshes stretch in certain directions (Fig. 2a). To address these issues, we extend affine transformations beyond simple similarity transformations to handle shear and stretch, which often occur in extreme poses. However, applying the same affine transformation to normals can lead to incorrect deformations, violating the orthogonality of their primitive-normals. To correct this, we introduce the *inverse-transpose* of the affine transformation. In previous similarity transformations, this step was unnecessary because rotations and scaling were handled separately, and the rotation matrix inherently preserved normal directions due to its invariance under the inverse-transpose operation.

Our method is fundamentally based on 3DMFM mesh binding inheritance, similar to GaussianAvatars (Qian et al., 2024), which uses 3DMFM-based mesh triangle deformation. However, if we only consider local deformation, discontinuities can occur between adjacent triangles (Zielonka et al., 2023b), and the method becomes limited in capturing deformations beyond the parametric head model. To address these discontinuities, previous methods (Zielonka et al., 2023b; Shao et al., 2024) blend the transformations of adjacent triangles using element-wise summation. However, as illustrated in Fig. 2b, this naive element-wise blending results in unnatural geometric interpolation within the matrix space. To overcome this issue, we propose the Jacobian Blend Skinning algorithm, which blends adjacent transformations while avoiding geometric distortions. This algorithm leverages the concept of linearizing the non-linear matrix interpolation space, drawing on classical matrix animation techniques (Shoemake & Duff, 1992) and employing geometrically smooth polar decomposition. Since the interpolation space of affine transformations is inherently non-linear, simple element-wise operations can lead to distortions (Fig. 2b).

Last but not least, we have witnessed a hollow illusion in the cornea, where a concave surface appears convex. This occurs during training as the model prioritizes photometric losses, constructing a concave retinal surface filled with low-bandwidth Spherical Harmonics (SHs) color to partially maintain specularity. However, this approach misleads the loss function by simulating mirror-like

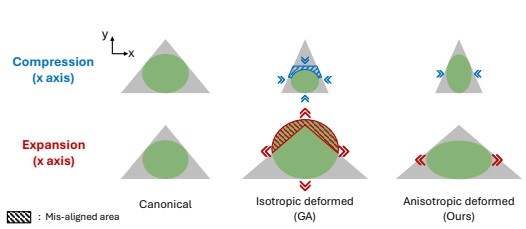

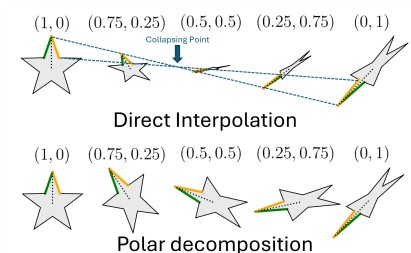

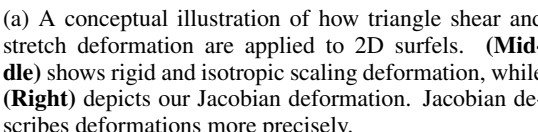

(a) A conceptual illustration of how triangle shear and stretch deformation are applied to 2D surfels. **(Middle)** shows rigid and isotropic scaling deformation, while **(Right)** depicts our Jacobian deformation. Jacobian describes deformations more precisely.

(b) Comparison of deformation interpolations between direct element-wise interpolation and polar decomposition interpolation. The above tuple label indicates the weight of each left and right-most matrix.

Figure 2: Toy examples on Jacobian deformation and Polar Decomposition.

specularity with SHs. To address this, we regularize corneal convexity and enhance specularity using computationally efficient Anisotropic Spherical Gaussians (ASGs) (Xu et al., 2013).

To summarize our key contributions,

- We introduce a novel representation for geometrically accurate 2D Gaussian primitive-based head avatars, utilizing intricate deformations driven by the affine Jacobian gradient instead of similarity transformation and corresponding normal adjustments.

- We propose a method called Jacobian Blend Skinning (JBS) to naturally interpolate affine transformations across adjacent deformations, effectively mitigating discontinuities, such as those arising from unseen pose extrapolation cases due to variations in a local space.

- We demonstrate the advantages of our methods across a variety of subjects captured with real and synthetic data, achieving superior results in challenging scenarios, such as sharp reflections on convex eyeballs, fine geometric details, and exaggerated deformations.

## 2 PROPOSED METHOD

This section highlights the key technical contributions of our work. We introduce a novel representation for geometrically accurate 2D Gaussian primitive-based head avatars, alongside a geometrically precise Jacobian deformation gradient and corresponding principled normal deformation (Sec. 2.2). Our proposed Jacobian Blend Skinning (JBS) approach (Sec. 2.3) allows for the interpolation of affine transformations, effectively addressing the discontinuities caused by piece-wise deformation in meshes. Additionally, we tackle the concavity (hollow-illusion) issue observed in the cornea by regularizing the modern 3DMFM model, FLAME (Li et al., 2017)'s eyeball model, to better manage high specularity through the use of Anisotropic Spherical Gaussians (ASGs) (Xu et al., 2013) (Sec. 2.4). We begin with preliminary on Gaussian Splatting and GaussianAvatars in Sec. 2.1.

### 2.1 PRELIMINARY

The two key building blocks of our approach are 2D Gaussian Splatting (2DGS) (Huang et al., 2024) and GaussianAvatars (Qian et al., 2024). The former provides intricate geometric properties, such as depth and normal. The latter employs a mesh-based rule-governed rigging strategy, allowing primitives to be rigged directly according to mesh triangles.

#### 2.1.1 GAUSSIAN SPLATTING

3D Gaussian Splatting (Kerbl et al., 2023) (3DGS) reconstructs a scene with anisotropic 3D Gaussian primitives. Each Gaussian is defined by a positive semi-definite covariance matrix $\Sigma$ that is centered at a position $\mu$. The covariance $\Sigma$ is decomposed as $\Sigma = RSS^T R^T$, where $R$ is a rotation matrix and $S$ is a diagonal scaling matrix. This covariance matrix is used when estimating each Gaussian's function value at 3D point $x$ such as $G(x) = \exp(-\frac{1}{2}(x - \mu)^T \Sigma^{-1}(x - \mu))$. Besides these geometrical properties, each Gaussian has appearance properties opacity $\alpha$ and color $c$. To render the scene, the final color $C$ is computed by alpha-blending Gaussians after projecting them to the image plane: $C = \sum_{i=1} c_i \alpha_i G^{proj}(x) \prod_{j=1}^{i-1}(1 - \alpha_i G^{proj}(x))$. $G^{proj}$ is given by evaluating

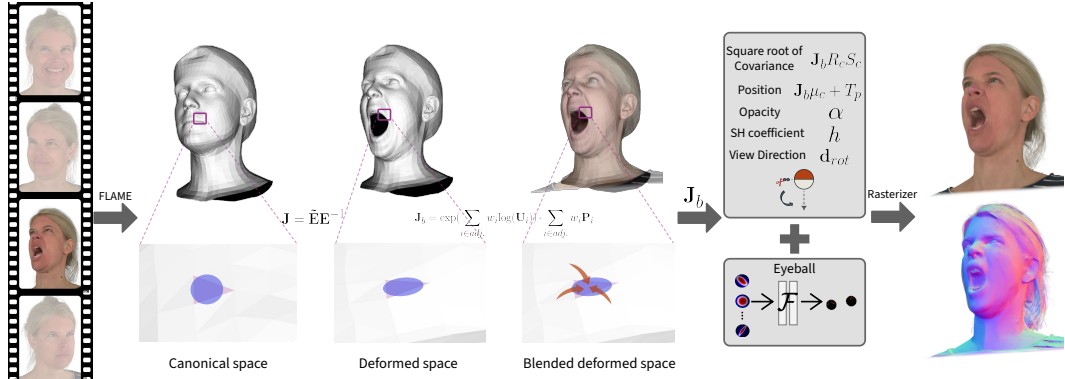

Figure 3: **Overall pipeline of SurFhead.** Only from RGB videos, **SurFhead** constructs geometrically accurate head avatars, equipped with our intricate deformations. The Jacobian $\mathbf{J}$ covers stretch and shear deformations avoiding surface distortion. Moreover, the blended Jacobian $\mathbf{J}_b$ alleviates inherent local deformations' discontinuity. Finally, elaborated modeling of eyeballs such as preservation of specularity and convexity achieves more realistic appearance and geometry.

the function value of 2D projection of the 3D Gaussian in image space by EWA volume splatting algorithm (Zwicker et al., 2001).

An extension to 3DGS, 2DGS (Huang et al., 2024), modifies 3D primitives to 2D "flat" surfels embedded in 3D space for reconstructing high-fidelity geometry. 2D surfels have some advantages compared with 3DGS in the respect of geometry; deriving intricate depth by ray-splat intersection algorithm (Weyrich et al., 2007), and closed-form modeling of the normal $n$ as cross-product of tangents $r_1$ and $r_2$. Thus, $R := [r_1; r_2; n]$, $n := r_1 \times r_2$, and each scales $s_1$, $s_2$, and 1 composes the scale matrix $S$. Details are provided in 2DGS (Huang et al., 2024) paper.

### 2.1.2 GAUSSIANAVATARS

GaussianAvatars (Qian et al., 2024) (GA) is an efficient head avatars model with FLAME (Li et al., 2017) mesh binding inheritance. Namely, each Gaussian has a single parent triangle in a canonical (local) space. When they are rendered in a deformed space, each Gaussian is transformed with their parent's current state such as relative rotation matrix $R_p$, relative triangle's area volume as isotropic scale $s_p$, and relative barycenter position of triangle $T_p$ in the world space. Namely,

$$R = R_p R_c, \mu = s_p R_p \mu_c + T_p, S = s_p S_c, \tag{1}$$

where $R_c$ is canonical rotations, $\mu_c$ is canonical position, and $S_c$ is canonical scaling in parent's triangle. Although this deformation approach is frequently utilized in certain head avatar research (Qian et al., 2024; Shao et al., 2024), we emphasize that it falls short in capturing the stretch and shear deformations that are essential for accurately extrapolating extreme expressions.

### 2.2 AFFINE TRANSFORMATION OF 2D SURFELS

Our goal is to reconstruct geometrically accurate head avatars using the 2D surfel splatting regime, which employs a normal consistency energy (Huang et al., 2024). This approach implies that high-fidelity normals originate from high-fidelity depths. Previous methods only adopt rigid deformation to maintain the positive semi-definite (PSD) property of Gaussian. However, as shown in Fig. 2a, using only rigid deformation for rigging can result in over- or under-occupying phenomena and deformation distortion.

**Local Geometry Descriptor.** Let us consider a triangle in the canonical space as the matrix formed by its vertices $v_i \in \mathbb{R}^3$, and its edge matrix as $\mathbf{E} = [v_1 - v_0, v_2 - v_0, v_3 - v_0] \in \mathbb{R}^{3 \times 3}$, where $v_3 := v_0 + \frac{(v_1 - v_0) \times (v_2 - v_0)}{\sqrt{|(v_1 - v_0) \times (v_2 - v_0)|}}$. $\tilde{\mathbf{E}}$ is the deformed version with the deformed triangle's vertices $\tilde{v}_i$ following Eq. 1. Note that in GaussianAvatars, the edge matrix is defined with a normalized base, height, and normal direction. This constructs an orthogonal matrix and approximates the shear or stretching-related deformations with isotropic scaling from the average of the base and height length. Please find a detailed derivation in GaussianAvatars (Qian et al., 2024).

**Jacobian Deformation Gradient for Surface.** The deformation gradient $\mathbf{J}$ is defined as $\tilde{\mathbf{E}}\mathbf{E}^{-1}$ following (Sumner & Popović, 2004). We introduce affine rigging with 2D surfels. The new parameterization is $\Sigma^{1/2} = \mathbf{J}R_c S_c$, instead of $s_p R_p R_c S_c$, where $\Sigma = \Sigma^{1/2}(\Sigma^{1/2})^T$. We note that the Jacobian deformation gradient $\mathbf{J}$ can correct inaccuracies in deformation, such as the lack of stretch or shear awareness, which can lead to incorrect surface reconstruction. It is also important to keep in mind that the Gaussian primitives retain their physical meaning only when the positive semi-definite is kept (Kerbl et al., 2023). We prove whether the covariance matrix $\Sigma$ remains PSD after applying affine transformations in Appendix A.2.2.

Both depth and normals are essential for constructing high-fidelity head geometry. While recent works on 3DGS-based full-body avatars (Zielonka et al., 2023a) and physical simulations (Xie et al., 2024; Jiang et al., 2024a) have shown that Jacobian deformation gradients accurately describe deformations, no prior work has addressed normal deformation with Jacobians. We found that the inverse-transpose of the Jacobian, $\mathbf{J}^{-T}$, is the correct deformation matrix for normals, based on the principle that "Orthogonality must be maintained in any space." In canonical space, if there is a tangent vector $r_c$ in a surfel, the normal $n_c$ is orthogonal to tangent ($n_c^T \cdot r_c = 0$). Then, the deformed tangent is defined as $r_d = \mathbf{J}r_c$. Let $\mathbf{A}$ be the matrix that transforms a normal vector from the canonical to the deformed space such as $n_d = \mathbf{A}n_c$. Trivially, the normal and tangent must preserve the orthogonality in the deformed space: $n_d^T \cdot r_d = (\mathbf{A}n_c)^T \cdot \mathbf{J}r_c = n_c^T \mathbf{A}^T \mathbf{J}r_c = 0$. We already know that $n_c^T r_c = 0$, then $\mathbf{A}^T = \mathbf{J}^{-1}$ (since $\mathbf{J}$ is non-singular). Therefore, $(\mathbf{A}^T)^T = \mathbf{A} = \mathbf{J}^{-T}$  □. Consequently, we use the principled transformation of normal as inverse-transposed form of Jacobian, $n_d = \mathbf{J}^{-T}n_c$.

### 2.3 Jacobian Blend Skinning (JBS)

Replacing the scaled-rotation component $s_p R_p$ of the similarity transform with the proposed $\mathbf{J}$ enables better representation of shear and stretch. However, ensuring the continuity of adjacent deformations is crucial for achieving smoother transformations. To address this, we introduce an improved technique to mitigate such discontinuities by substituting $\mathbf{J}$ again with a blended Jacobian, $\mathbf{J}_b$. We refer to this approach as Jacobian Blend Skinning (JBS), which builds on the principles of Linear Blend Skinning (LBS) (Badler & Morris, 1982) while addressing its inherent limitations.

**Degenerate Solution of Linear Blend Skinning in Matrix Space.** Linear Blend Skinning (LBS) is widely used for dynamic modeling of the human body and head (Li et al., 2017; Loper et al., 2015), but it struggles with the non-linearity of rotation matrix interpolation in $SO(3)$, causing distortions in rotational transformations. Our toy experiments (Fig. 2b) show how element-wise linear interpolation distorts shapes. Previous works (Zielonka et al., 2023b; Shao et al., 2024) also adopt this sub-optimal approach to handle local deformations. Techniques like matrix-logarithm interpolation or quaternion's SLERP can address this but are limited to rotations.

**Introducing Jacobian Blend Skinning (JBS).** The Jacobian Blend Skinning (JBS) algorithm overcomes the limitations of LBS by focusing on the interpolation of Jacobian gradients, which encapsulate not only rotations but also shear and stretch, both residing in the broader $GL(3)$ (General Linear) field. The key to JBS is leveraging Polar Decomposition (PD) (Shoemake & Duff, 1992) to break down Jacobian gradient $\mathbf{J}$ into two meaningful components: rotation (orthogonal matrix $\mathbf{U}$) and stretch/shear (symmetric positive semi-definite matrix $\mathbf{P}$): $\mathbf{J} = \mathbf{UP}$. This decomposition is unique and coordinate-independent (proof in Appendix A.2.1), making it geometrically sound for interpolation. By blending the rotation in matrix-logarithm space and the stretch/shear in linear matrix space, JBS ensures that the resulting transformations remain geometrically valid, avoiding the distortions seen with LBS (Fig. 2b).

JBS is applied to the Gaussian deformations considering not only its parents' triangle, but its adjacent triangles. The blending itself is performed separately for the rotational ($\mathbf{U}_b$) and stretch/shear ($\mathbf{P}_b$) components. For rotations, the blending weights are applied in the matrix-logarithm space $\mathfrak{so}(3)$, followed by an exponential mapping back into the $SO(3)$ space. For stretch/shear, the weights are applied directly in the linear matrix space, ensuring positive semi-definiteness, which avoids geometrical distortions. Mathematically, this Jacobian Blend Skinning is defined as:

$$\mathbf{J}_b := \text{JBS}(\mathbf{J}, w) = \underbrace{\exp\left(\sum_{i \in adj.} w_i \log(\mathbf{U}_i)\right)}_{\mathbf{U}_b} \cdot \underbrace{\sum_{i \in adj.} w_i \mathbf{P}_i}_{\mathbf{P}_b}, \qquad (2)$$

where $adj.$ implies the set of adjacent triangles. The blending weights, $w_i$, in JBS are learnable convex weights that control how much influence each adjacent triangle has on the resulting blended Jacobian $\mathbf{J}_b$. Note that since $w_i$ is activated by Sigmoid function, the blended stretch/shear component $\mathbf{P}_i$ is also guaranteed positive semi-definite property. The exponential and logarithm mappings for rotations are performed using Rodrigues' formula. Finally, the blended Jacobian $\mathbf{J}_b$ is then used to deform the canonical Gaussian's covariance and mean position and also transform the normal.

$$\Sigma^{1/2} = \mathbf{J}_b R_c S_c, \mu = \mathbf{J}_b \mu_c + T_p \tag{3}$$

$$n_d = \mathbf{J}_b^{-T} n_c \tag{4}$$

In summary, JBS improves on LBS by using Polar Decomposition to separately blend rotation and stretch/shear transformations. This approach ensures more accurate and geometrically meaningful deformations, avoiding the issues caused by linear interpolation. By operating in the appropriate spaces—matrix-logarithm for rotations and linear for stretch/shear—JBS preserves the integrity of transformations, making it a more reliable solution for complex deformations.

## 2.4 RESOLVING HOLLOW-ILLUSION IN EYEBALLS

Many studies (Park et al., 2021; Li et al., 2022; 2024) have witnessed that volumetric approaches often fail to produce a good approximation of the eyeball geometry and yield hollow eyes. We also have empirically found that the cornea of the human eye often appears as concave geometry without proper constriction. Since the cornea exhibits high specularity due to its multiple membranes (Dua et al., 2013), the limited representation capability Spherical Harmonics (SHs) can distort the geometry, unnaturally struggling to satisfy the photometric losses. This deceptive phenomenon is harmful to achieving accurate geometric representation as can be seen in the inset.

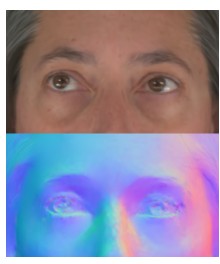

To address the issue of concave geometry around the eyeball, we eliminate the geometrical gradient on the eyeball-bound Gaussians, excluding Gaussians' cloning and splitting. We use the FLAME eyeball mesh as an approximation for the eyeball geometry. Additionally, we regularize the opacity of the eyes to approach 1, following the energy $\mathcal{L}_{eye}$ in Sec. 3. Although this improves the geometry, SHs still fall short of capturing eye specularity.

In mitigation, we employ Anisotropic Spherical Gaussians (ASGs) (Xu et al., 2013). Since the cornea and sclera often reflect their light environment, ASGs excel at capturing sharp reflections. Implementation details of ASGs can be found in Appendix A.3. To preserve computational efficiency, we leverage common knowledge from casual data capture. First, eyeballs aren't affected by light from the back of the head, so we limit the sampling range to the frontal hemisphere of the world space, reducing ASGs by 50% compared to previous methods (Han & Xiang, 2023). Second, given that captured data is often in environments with ample white light, representing specular reflections as a monochrome intensity channel is sufficient.

## 3 TRAINING STRATEGY

### 3.1 OPTIMIZATION

We supervise the rendered images with photometric loss $\mathcal{L}_{photo}$ which is a combination of L1 term $\mathcal{L}_{l1}$ and a D-SSIM term $\mathcal{L}_{ssim}$ following 3DGS (Kerbl et al., 2023). Moreover, since we aim to reconstruct high-fidelity geometry, we followed the geometric energies in 2DGS (Huang et al., 2024), depth-distortion $\mathcal{L}_{depth}$ and normal consistency energies $\mathcal{L}_{normal}$. Toward the better alignment between Gaussians and their parent triangles, we utilize two regularization energy terms $\mathcal{L}_{scaling}$ and $\mathcal{L}_{position}$ from GA (Qian et al., 2024).

**Eyeball Regularization.** We have empirically found that the highly specular eyeball region, especially the cornea, tends to yield overly transparent Gaussians to satisfy the photometric losses. This leads to an incorrectly shaped, concave cornea, although the genuine surface should be roughly convex (Li et al., 2022). To alleviate this issue, we constrain the cornea regions to be opaque by regularizing the opacity of the respective Gaussians: $\mathcal{L}_{eye} = \sum_{i \in E} (1 - \alpha_i)^2$, where the $E$ denotes the set of Gaussian in the eyeball region. Since we utilize the 3DMFM parametric mesh, this

set can be easily extracted. We provide more details about the entire energies with coefficients in Appendix A.3.

## 3.2 ADAPTIVE DENSITY CONTROL

We enable adaptive density control with binding inheritance (Qian et al., 2024). Moreover, to avoid degenerate transparent solutions of eyeballs, we stop the gradient of eyeball-bound Gaussians' rotation $R_c$ and position $\mu_c$ by blocking the proliferation, initialized identity and zero. We describe the proliferation of Gaussians, so-called Adaptive Density Control (ADC) (Kerbl et al., 2023) below.

**Occlusion Gradient Amplification.** The original adaptive density control (ADC) strategy (Kerbl et al., 2023) relies on view-space gradients. Occluded regions such as the lower tooth tend to suffer from less frequent updates. To remedy this, we amplify the view-space gradients of the tooth by $20\times$. This amplifier is a hyper-parameter that depends on the extent of the occluded parts shown.

## 4 EXPERIMENTS

As shown in Fig. 3, the input to our pipeline is a tracked RGB video. For head tracking, we adopt the same preprocessing approach used in GaussianAvatars (Qian et al., 2024) (GA) for the multiview RGB video dataset, NeRSemble (Kirschstein et al., 2023). Furthermore, since the strength of our method lies in accurately reconstructing the geometry of dynamic head avatars, we validate our approach using a synthetic dataset FaceTalk (Zheng et al., 2022) that includes ground truth normal.

### 4.1 EVALUATION PROTOCOL

#### 4.1.1 BASELINES

For evaluation, we train our model with five baselines: IMAvatar (Zheng et al., 2022), FLARE (Bharadwaj et al., 2023), PointAvatar (Zheng et al., 2023), SplattingAvatar (Shao et al., 2024), and GaussianAvatars (Qian et al., 2024). IMAvatar employs a neural occupancy field, FLARE uses mesh-based avatars with intrinsic material decomposition, and PointAvatar adopts a point-based explicit model. GaussianAvatars and SplattingAvatars, utilizing mesh-binding inheritance, represent 3D Gaussian splatting. We exclude 3D Gaussian splatting models for the synthetic dataset due to the lack of surface normals and omit IMAvatar for the real dataset due to training instability.

#### 4.1.2 DATASETS

**Monocular Synthetic Dataset (FaceTalk (Zheng et al., 2022)).** FaceTalk, rendered from the FLAME (Li et al., 2017) model, includes diverse expressions and poses at a resolution of $512 \times 512$. With ground truth normals available, we assess geometry fidelity. For experiments, we selected five identities, using 49 sequences for training and 3 with extreme poses for testing.

**Multi-view Real Dataset (NeRSemble (Kirschstein et al., 2023)).** NeRSemble, a real human head dataset with 16 cameras, follows the protocol of GaussianAvatars, using 11 video sequences (four emotions, six expressions, and one free performance). One sequence is reserved for testing, and the free performance sequence is used for cross-identity reenactment. To assess generalization for extreme poses, we manually constructed a new held-out set (detailed in Appendix A.3). LPIPS (Zhang et al., 2018) is excluded for efficiency but included in Tab. 2 only for fair comparison.

#### 4.1.3 METRICS AND TASKS

To evaluate self-reenactment and dynamic novel-view synthesis, we use PSNR, SSIM, perceptual LPIPS (Zhang et al., 2018), and normal cosine similarity (NCS). As the FaceTalk dataset (Zheng et al., 2022) lacks multi-view data, we only evaluate NVS on the NeRSemble dataset (Kirschstein et al., 2023). For NCS on NeRSemble, we use pseudo-ground truth normals from the Sapiens model (Khirodkar et al., 2024). Alongside quantitative evaluations, we report qualitative cross-reenactment results in Fig. 5. Each table highlights Best and Second best scores.

Table 1: Quantitative comparison on FaceTalk (Zheng et al., 2022).

|  | PSNR↑ | SSIM↑ | LPIPS↓ | NCS↑ |
|---|---|---|---|---|
| IMAvatar | 32.23 | 0.983 | 0.037 | 0.931 |
| PointAvatar | 34.93 | 0.984 | 0.065 | 0.756 |
| FLARE | 31.64 | 0.968 | 0.027 | 0.943 |
| **Ours** | 40.15 | 0.992 | 0.020 | 0.983 |

Table 2: Quantitative comparison on NeRSemble (Kirschstein et al., 2023).

|  | Novel-View Synthesis | | | | Self-Reenactment | | | |
|---|---|---|---|---|---|---|---|---|
|  | PSNR↑ | SSIM↑ | LPIPS↓ | NCS↑ | PSNR↑ | SSIM↑ | LPIPS↓ | NCS↑ |
| PointAvatar | 20.56 | 0.844 | 0.206 | 0.410 | 20.59 | 0.854 | 0.190 | 0.611 |
| Flare | 21.91 | 0.814 | 0.228 | 0.817 | 21.11 | 0.802 | 0.227 | 0.737 |
| SplattingAvatars | 23.68 | 0.858 | 0.232 | 0.704 | 20.25 | 0.828 | 0.265 | 0.688 |
| GaussianAvatars | 30.29 | 0.934 | 0.067 | 0.797 | 23.43 | 0.891 | 0.093 | 0.716 |
| **Ours** | 30.07 | 0.934 | 0.079 | 0.896 | 23.53 | 0.892 | 0.103 | 0.832 |
| Ours + LPIPS | 29.94 | 0.933 | 0.062 | 0.894 | 23.78 | 0.894 | 0.089 | 0.826 |

## 4.2 HEAD AVATAR RECONSTRUCTION AND REENACTMENT

**Synthetic Dataset.** Fig. 4 presents qualitative comparisons on FaceTalk synthetic dataset. IMAvatar (Zheng et al., 2022) demonstrates plausible rendering and geometry, but it often misses geometrical details with oversmoothing, particularly in the ears and nasal line, and displays concave artifacts in the pupil region. Furthermore, this method is quite slow, as training requires numerical searches to locate the surface, making it approximately 200× slower than PointAvatar (Zheng et al., 2023). While PointAvatar offers faster performance, it suffers from dotted noise in geometry due to its fixed, isotropic point size, making it challenging to represent extreme poses. FLARE (Bharadwaj et al., 2023) also shows mangled rendering and geometry, particularly in cases of extreme facial expressions, as it involves decomposing intrinsic material from a single environment, an ill-posed problem. This trend is reflected in the quantitatives in Tab. 1.

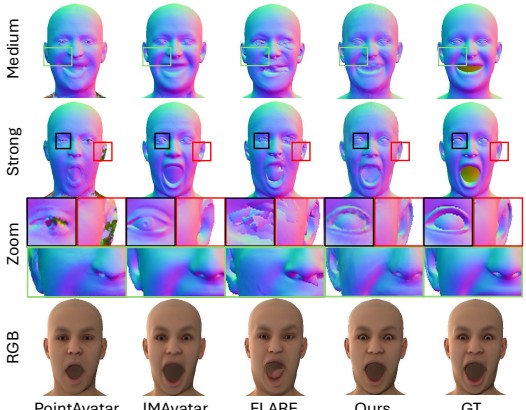

Figure 4: Qualitative results on the FaceTalk dataset (Zheng et al., 2022). Distinguished from other baselines, **SurFhead** simultaneously achieves convex eyeballs, and highly-detailed ears and nasal line. Best viewed when zoomed in.

**Real Dataset.** For the NeRSemble real dataset, Fig. 5 shows qualitative comparisons. FLARE (Bharadwaj et al., 2023)'s mesh involving unstable remeshing technique leads both oversmoothed rendering and normal with inferior quality. PointAvatar (Zheng et al., 2023) excels in fine detailed normals such as wrinkles, owing to their drastic pruning strategy to avoid the ambiguity of normal volumetric rendering, but they show salt-and-pepper artifacts from inflexible point representation. We also regard these two baselines' degradations stem from their inherent representations intractability of their personalized blendshape space. SplattingAvatar (SA) (Shao et al., 2024) and GaussianAvatars (GA) (Qian et al., 2024), which utilize 3DGS (Kerbl et al., 2023), aim for high-quality renderings. Note that the normal from SA and GA is derived from the shortest axis of the Gaussians, a method commonly used in recent relighting research (Jiang et al., 2024b). SA lacks explicit regularization of Gaussian positions, allowing them to drift far from their parent triangles, leading to inferior normal quality and floating artifacts. In both quantitative (Tab. 2) and qualitative evaluations, GA produces results comparable to ours, but its coarse deformation strategy, similarity transformation, does not account for triangle stretching and deformation discontinuity, resulting in semi-transparent and blob-like artifacts during extreme pose extrapolation. Additionally, SA and GA's normal quality suffer due to the absence of geometry-level optimizations. Otherwise, thanks to our design of capturing high-fidelity geometry and accurate deformations, ours show superior quality of normal with detailed geometry. Not only the geometry, ours outperforms other state-of-the-art methods in terms of rendering quality. The qualitative observation is also proved in Tab. 2. Notably, GA shows better PSNR and LPIPS than ours, owing to 3DGS (Kerbl et al., 2023)'s representations. This phenomenon in rendering quality gap is also reported in 2DGS (Huang et al., 2024) in view of the absence of representation dimensionality. Although this numerical scores, as can be seen in red boxes Fig. 5, ours result is more robust than GA on extreme expression scenarios. However, ours with LPIPS achieves the best LPIPS with preserving other metrics. Ours scores best in the NCS which indicate its high fidelity for dynamic geometry reconstruction with a large margin. In summary, ours outperform other baselines in terms of reconstruction capability with appearance and geometry, both.

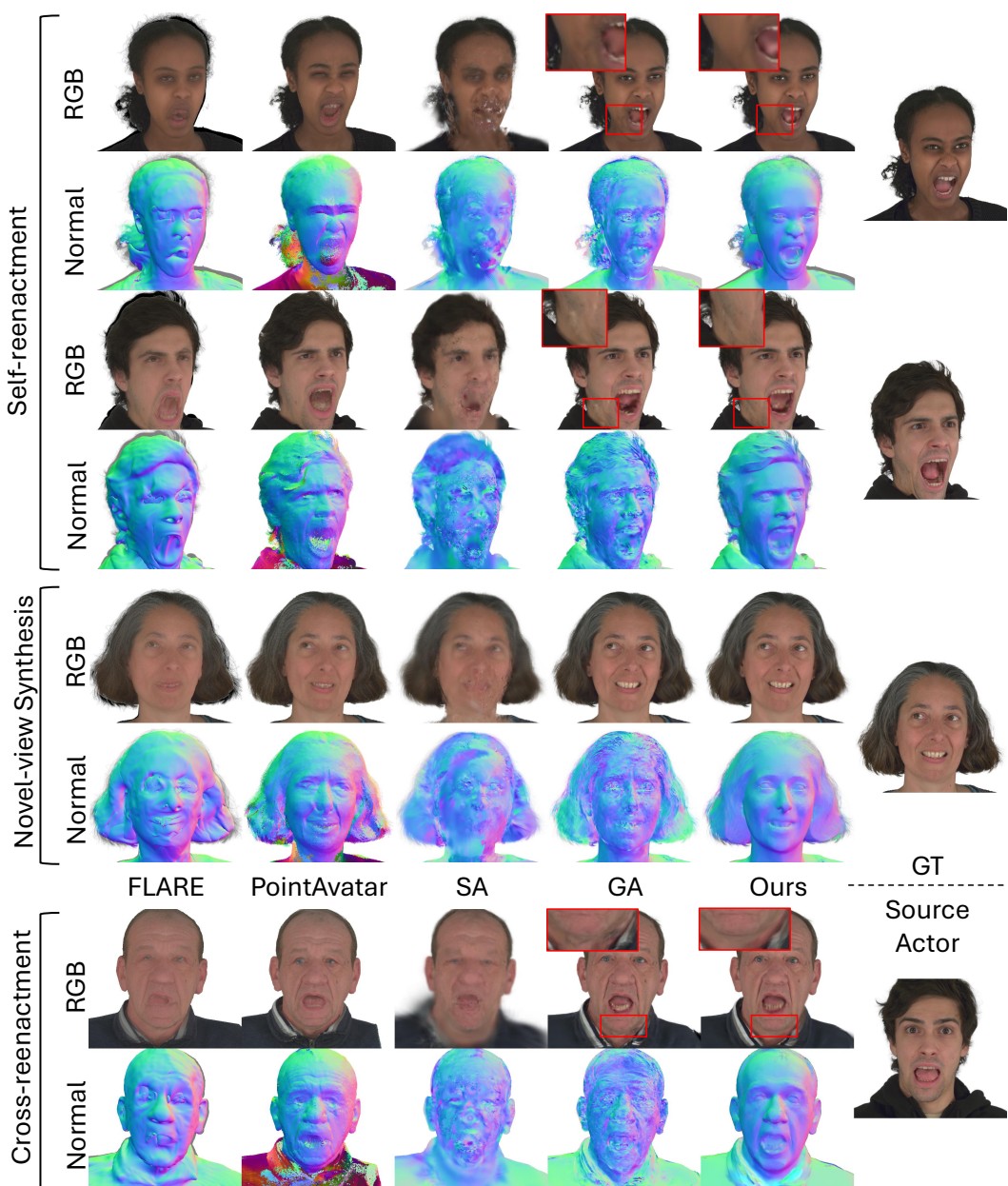

Figure 5: Qualitative results on NeRSemble dataset (Kirschstein et al., 2023). Thanks to Jacobian and their blending, our method produces high-quality geometry with intricate details, visible in the normal maps. Please be aware of red boxes.

## 4.3 ABLATION STUDY

**Jacobian and JBS help intricate rigging.**

We further validate the effectiveness of our method's Jacobian and Jacobian Blend Skinning (JBS) in Tab. 3 and Fig. 6a. Here, "Vanilla" refers to the combination of 2DGS (Huang et al., 2024) and Gaussian Avatars (Qian et al., 2024).

When incorporating the Jacobian deformation gradient, we observe a noticeable improvement in normal quality, particularly in reducing artifacts near the jaw and nasal lines, resulting in more coherent normal representations.

Table 3: Ablations. +/- indicate incremental changes from the previous.

| | PSNR↑ | SSIM↑ | LPIPS↓ | NCS ↑ |
|---|---|---|---|---|
| GaussianAvatars | 22.49 | 0.920 | 0.089 | 0.727 |
| Vanilla | 22.32 | 0.907 | 0.093 | 0.803 |
| + eyeballs | 22.35 | 0.901 | 0.093 | 0.809 |
| + Jacobian | 22.38 | 0.902 | 0.091 | 0.812 |
| + JBS (= Ours) | 23.09 | 0.931 | 0.082 | 0.845 |
| - ASGs | 23.07 | 0.922 | 0.089 | 0.846 |
| - eyeballs | 23.03 | 0.925 | 0.087 | 0.820 |

(a) Jacobian and JBS help to reduce render artifacts and messy normal with extreme poses.

(b) Both ASGs and eyeball regularization are needed for accurate geometry and sharp specular rendering.

Figure 6: Qualitative ablation for proving the proposed method. The left side show how Jacobian deformations and Jacobian Blend Skinning alleviate artifacts in extreme poses. The right side shows degradations in the eye region when training without eyeballs or without anisotropic spherical Gaussians (ASG).

Rendered images show better alignment between floating artifacts and surface geometry, creating a more structured and accurate overall appearance. With JBS applied, the normal achieves even higher quality, showing smoother and more distinct separation. Pop-out artifacts are nearly eliminated, leading to refined representations of facial features, especially around the nasal area, surpassing the coarse expressions captured by the FLAME mesh. JBS significantly enhances the fidelity and detail of the normal maps by compensating for deformation discontinuities.

**Effects of ASGs and Eyeball Regularization.** To model the cornea of the eyeballs more accurately, we propose the use of Anisotropic Spherical Gaussians (ASGs) with a new, efficient implementation and eyeball regularization. To validate these improvements, we present qualitative results in Fig. 6b and quantitative results in Tab. 3. Without ASGs, the rendered color of the eyeball regions shows low-frequency reflections, resulting in a matte appearance. This occurs because, during the optimization of the eyeball region with convexity regularization, Spherical Harmonics alone are insufficient to capture the specular highlights. Moreover, when eyeball regularization is further reduced, the eyeballs appear to retain specularity, but the normals exhibit concave shapes, creating a hollow illusion. These observations suggest that the intricate modeling of both the geometry and appearance of the eyeballs is not feasible without either our ASGs or eyeballs regularization.

## 5 CONCLUSION AND DISCUSSION

**SurFhead** introduces a novel method for reconstructing dynamic head avatars that strikes a balance between photorealism and geometrically accurate rigging. By integrating Jacobian deformation with detailed normal adjustments and Jacobian Blend Skinning (JBS), our approach enables precise control over both appearance and geometry. As a result, SurFhead surpasses existing state-of-the-art dynamic head models, excelling in accurate geometry reconstruction, pose and expression extrapolation, and demonstrating its applicability in areas such as relighting (see Appendix A.6) and dynamic mesh reconstruction by leveraging the efficient, rule-governed rigging regime of 3DMFM meshes.

However, there is still room for improvement in the near future. First, similar to other 3DMFM and Gaussian-based methods, certain challenges persist, particularly regarding the **bounded representation of the expression space in 3DMFM**. Additionally, expressions that are grossly exaggerated and fall outside the span of 3DMFM's expression space, even cannot be head tracked in the preprocessing stage. Furthermore, elements like the tongue and individual hair strands are still missing in modern 3DMFMs. Besides, one of the strengths of our rule-governed deformation from 3DMFM meshes is efficiency. Therefore, we also discuss the improvement room for **computational efficiency of polar decomposition**, which, unlike other Gaussian primitive-based methods relying on black-box learning, is deterministically governed in SurFhead. These further discussions on these limitations are in Appendix A.6.

## ACKNOWLEDGMENT

This research was supported by the Institute for Information & Communications Technology Planning & Evaluation (IITP) grant, funded by the Korean government (MSIT) (RS-2019-II190075, Artificial Intelligence Graduate School Program at KAIST). We also acknowledge the support of the National Research Foundation of Korea (NRF) grant, funded by the Korean government (MSIT) (No. RS-2025-00555621). Additionally, we would like to express our sincere gratitude to Jungwoo Ahn for conducting the baseline experiments.

## REPRODUCIBILITY STATEMENT

To ensure the reproducibility of our method, we provide the anonymous GitHub link. Moreover, the details on implementation and the information of datasets can be found in the appendix A.3 and Sec. 4.1.2, respectively. For NeRSemble (Kirschstein et al., 2023) dataset, we used pre-processed dataset from GaussianAvatars (Qian et al., 2024). Together with provided code and publicly available datasets, we believe our paper contains sufficient information for reimplementation.

## ETHICS STATEMENT

The creation of highly accurate and geometrically detailed head avatars raises significant ethical and security concerns, particularly regarding privacy violations. The realistic nature of these avatars can lead to unauthorized manipulation, where personal likenesses may be misused without consent. This includes altering appearances in virtual environments or generating misleading content, which infringes on privacy and may result in serious consequences.

One critical issue is the potential for creating deepfake videos, which could spread false information, harm reputations, and manipulate public opinion. The increased realism of these avatars makes it difficult to discern authentic content from fabricated material, amplifying risks of misinformation and defamation. Additionally, the possibility of identity theft through the exploitation of digital avatars presents a serious threat to personal security and public trust in digital media.

To mitigate these risks, we will strictly limit the use of our method to academic research. This restriction aims to prevent misuse and ensure the technology is applied in an ethical and constructive manner.

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

# A   APPENDIX

## A.1   RELATED WORK

**Static to Dynamic Radiance Field Reconstruction.** NeRF (Mildenhall et al., 2021) and NV (Lombardi et al., 2019) have opened the era of photorealistic renderings with undertaking novel-view synthesis task. Optimization and rendering efficiency could be improved by hash encoding (Müller et al., 2022) and tensor decomposition (Chen et al., 2022). Later, Mixture of Volumetric Primitives (Lombardi et al., 2021) proposes efficient methodology with surface-aligned cuboid primitives which boost only ray marching around surfaces. Few years later, 3D Gaussian Splatting (3DGS) (Kerbl et al., 2023) utilizes anisotropic 3D Gaussians by rasterizing them without expensive ray-marching like previous strategies. We leverage the 3DGS, benefiting from the expressiveness and efficiency.

**Neural Surface Reconstruction.** Emerging from the success of NeRF (Mildenhall et al., 2021), neural surface reconstruction (Oechsle et al., 2021; Yariv et al., 2020; Wang et al., 2021) has garnered significant attention from researchers, leveraging Occupancy Fields and Signed-Distance Functions (SDFs). Notably, subsequent works (Rosu & Behnke, 2023; Li et al., 2023) have introduced methodologies that integrate fast and efficient hash-encoding-based radiance fields (Müller et al., 2022) with SDFs. This development has inspired research into human head geometry reconstruction, moving beyond mesh-based topology-invariant 3D Morphable Face Models (3DMFMs) (Li et al., 2017; Paysan et al., 2009). The first major work in this area, H3D-Net (Ramon et al., 2021), proposed an SDF-based approach for few-shot human head geometry reconstruction using an auto-decoder-trained model. Building on this, a subsequent study (Xu et al., 2023a) targeted a similar setting as H3D-Net, focusing on more detailed human head geometry reconstruction with single-person refinement. Recently, MonoNPHM (Giebenhain et al., 2024a) introduced a few-shot head reconstruction framework based on a pretrained neural parametric head model (Giebenhain et al., 2023), which combines the topology-free advantages of SDFs with the manipulability of parametric models. Departing from these SDF-based approaches, we are the first to propose reconstructing dynamic head geometry from RGB videos, including monocular setups, utilizing Gaussian primitives (Huang et al., 2024).

**Reconstructing and Animating Head Avatars.** Existing approaches for reconstructing and animating avatars mainly differ in two fundamental aspects: implicit or explicit models. Implicit models reconstruct the face by neural radiance field in combination with volumetric rendering or using implicit surface functions. (Gao et al., 2022), (Zielonka et al., 2023b), (Zheng et al., 2022), (Xu et al., 2023b) With explicit models (Grassal et al., 2022),(Zheng et al., 2023),(Khakhulin et al., 2022), the seminal work of 3DMFM uses principal component analysis (PCA) to model facial appearance and geometry on a low-dimensional linear subspace. 3DMFM and its variants have been widely applied in optimization-based and deep learning-based head avatar creation. Recently, due to its efficient rendering and topological flexibility, there are many works utilizing 3D Gaussian Splatting (Kerbl et al., 2023). These works can be further categorized based on how they define and use Gaussians. Some approaches bind Gaussians directly to the mesh (Shao et al., 2024; Qian et al., 2024), while others map them onto UV coordinates (Xiang et al., 2024). In some cases, Gaussians are extracted as features (Giebenhain et al., 2024b),(Xu et al., 2024), or a neural parametric model replaces the traditional 3DMFM (Giebenhain et al., 2024b). However, in most cases, the deformation is often handled by the mesh itself with little additional consideration.

## A.2   PROOFS

### A.2.1   POLAR DECOMPOSITION'S UNIQUENESS

*Lemma 1.* There exists a unique polar decomposition of an arbitrary invertible matrix $\mathbf{A}$ such that

$$\mathbf{A} = \mathbf{UP},$$

where $\mathbf{U}$ is an orthogonal matrix and $\mathbf{P}$ is a positive semidefinite symmetric matrix.

*Proof (Lemma 1).* Suppose that there are two polar decompositions of the matrix $\mathbf{A}$:

$$\mathbf{A} = \mathbf{U}_1\mathbf{P}_1 = \mathbf{U}_2\mathbf{P}_2,$$

where $\mathbf{U}_1, \mathbf{U}_2$ are orthogonal matrices and $\mathbf{P}_1, \mathbf{P}_2$ are positive semidefinite symmetric matrices.

Now, multiply both sides of $\mathbf{U}_1\mathbf{P}_1 = \mathbf{U}_2\mathbf{P}_2$ by $\mathbf{U}_1^T$ on the left:

$$\mathbf{P}_1 = \mathbf{U}_1^T\mathbf{U}_2\mathbf{P}_2.$$

Let $\mathbf{Q} = \mathbf{U}_1^T\mathbf{U}_2$, so that

$$\mathbf{P}_1 = \mathbf{Q}\mathbf{P}_2.$$

Since $\mathbf{Q}$ is an orthogonal matrix and both $\mathbf{P}_1$ and $\mathbf{P}_2$ are positive semidefinite symmetric matrices, we conclude that $\mathbf{Q} = \mathbf{I}$, the identity matrix. Hence,

$$\mathbf{U}_1^T\mathbf{U}_2 = \mathbf{I} \quad \Rightarrow \quad \mathbf{U}_1 = \mathbf{U}_2.$$

Thus, $\mathbf{P}_1 = \mathbf{P}_2$, and therefore the polar decomposition $\mathbf{A} = \mathbf{U}\mathbf{P}$ is unique. $\square$

### A.2.2 PRESERVATION OF POSITIVE SEMIDEFINITE WITH JACOBIAN GRADIENTS

*Lemma 2.* Consider an arbitrary matrix $M$. The matrix multiplication $MM^T$ is positive semidefinite (PSD) because for any vector $\mathbf{x}$,

$$\mathbf{x}^T MM^T\mathbf{x} \geq 0 \Leftrightarrow \left\|M^T\mathbf{x}\right\|^2 \geq 0.$$

The matrix $\Sigma$ also follows this form. Therefore, $\Sigma$ retains its positive semidefinite property after the affine transformation. $\square$

### A.3 IMPLEMENTATION DETAILS

**Optimization Specifications.** We use Adam (Kingma & Ba, 2014) optimizer for learnable Gaussian parameters and translation, joint rotations, and expression parameters of FLAME (Li et al., 2017). We set the learning rates for all parameters same as with GA (Qian et al., 2024), except for the blending weights $w$. For the $w$, we set the learning rate as 1e-3. We train for 300,000 iterations, and exponentially decay the learning rate for the $\mu$ until the final iteration, where it reaches $0.01\times$ the initial.

Finally, the entire energies are defined as:

$$\mathcal{L} = \mathcal{L}_{photo} + \lambda_{\text{depth}}\mathcal{L}_{depth} + \lambda_{\text{normal}}\mathcal{L}_{normal} + \lambda_{\text{eye}}\mathcal{L}_{eye}. \tag{5}$$

The energy balances are $\lambda_{depth} = 100$, $\lambda_{normal} = 0.05$, and $\lambda_{eye} = 0.1$.

**Calculation Jacobian with Covariance.** Since there is no support for precomputation of covariance with original 2DGS (Huang et al., 2024) rasterizer implementation [1], we calculate the Jacobian deformation gradient $\mathbf{J}_b$ combining with 3DGS's original covariance as NVIDIA CUDA kernel implementation.

**Color Change with Geometrical Transformations.**

Most previous dynamic head avatars almost neglect the color change with deformations (Zielonka et al., 2023b; Qian et al., 2024; Shao et al., 2024; Lombardi et al., 2021). When the bases of SHs do not rotate, they just query the same direction with facing identical side of Gaussians in any deformed space. This phenomenon is illustrated in the inset of the upper right branch. To mitigate this, we suggest a simple solution like the lower right of the inset, we inversely rotate the view direction $\mathbf{d}$ with the inverse of Gaussian's blended rotation part $\mathbf{U}_b^T$. Namely, we rewrite the rotated input view direction: $\mathbf{d}_{\text{rot}} = \mathbf{U}_b^T\mathbf{d}$.

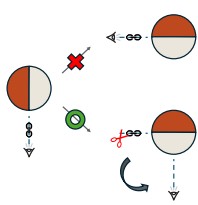

**Anisotropic Spherical Gaussians for Eyeballs' specularity.** Comparing with Spherical Gaussians (SGs), ASGs (Xu et al., 2013) have been demonstrated to effectively represent anisotropic scene with a relatively small number.

ASGs are theoretically defined as:

$$ASG(\nu|[x,y,z],[\lambda,\mu],\xi) = \xi \cdot max(\nu \cdot z, 0) \cdot e^{-\lambda(\nu \cdot x)^2 - \mu(\nu \cdot y)^2}, \tag{6}$$

---

[1]https://github.com/hbb1/diff-surfel-rasterization.git

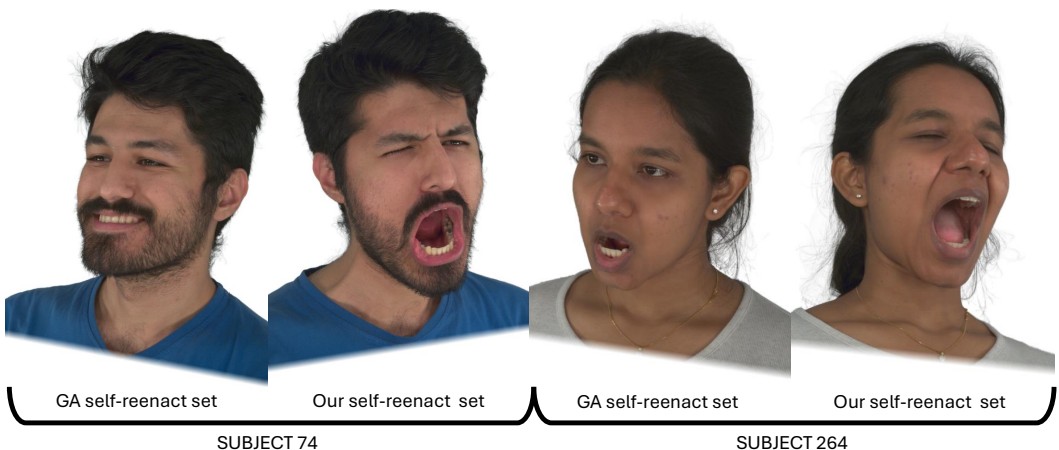

| | | | |
|---|---|---|---|
| GA self-reenact set | Our self-reenact set | GA self-reenact set | Our self-reenact set |
| SUBJECT 74 | | SUBJECT 264 | |

Figure 7: Dataset example of custom self-reenactment held-out.

| Subject ID | 074 | 140 | 175 | 210 | 253 | 264 | 302 | 304 | 306 |
|---|---|---|---|---|---|---|---|---|---|

Table 4: The held-out sequence of each subject for self-reenactment evaluation.

where z is lobe-axis, x and y are each tangent and bi-tangent of z, $\{\lambda, \mu\} \in \mathbb{R}^+$ are sharpness parameters, $\nu$ is the unit direction function input, and the $max$ term implies the smooth term.

Given that most specular BRDFs feature a lobe aligned with a specific reflection direction, we calculate the reflection vector to serve as the input direction: $\boldsymbol{\omega}_o = 2(\mathbf{d}_{\rm rot} \cdot n)n - \mathbf{d}_{\rm rot}$, where $\mathbf{d}_{\rm rot}$ is the rotated input view direction, $n$ is a normal direction computed for each Gaussian, and $\boldsymbol{\omega}_o$ is the reflect direction. Lastly, since a simple summation of ASG output limits its representative ability, we utilize tiny two-hidden layered MLP $\mathcal{F}$ to obtain the final specular color, following (Yang et al., 2024):

$$c_s(\boldsymbol{\omega}_o; \mathbf{x}) = \mathcal{F}\left( \bigoplus_{i=0}^{N-1} ASG(\omega_o | [\boldsymbol{\omega}_i^\lambda, \boldsymbol{\omega}_i^\mu, \boldsymbol{\omega}_i], [\lambda_i, \mu_i], \xi_i), \gamma(\mathbf{d}_{\rm rot}), \mathbf{n} \cdot \mathbf{d}_{\rm rot} \right) \tag{7}$$

where $\bigoplus$ indicates the concatenation operation, $\gamma$ denotes the positional encoding, and N is the number of basis SGs.

Not to compromise the computational efficiency, we leverage the common prior knowledge from the casual data capture setting. First, the eyeballs could not be effected by lights from backside of heads. Therefore, we sample $\boldsymbol{\omega}_i$ with limited range from frontal hemisphere of world space; $N = 4 \times 4$ relatively $50\%$ reduced ASGs compared with (Han & Xiang, 2023; Yang et al., 2024). Second, since the captured data typically lie under environments with abundant white light, representing the MLP $\mathcal{F}$ output color as a monochrome channel for intensity is sufficient.

For a sampled direction $\boldsymbol{\omega}_i = (\theta, \phi)$ in the spherical coordinate system, we set $\boldsymbol{\omega}_i^\lambda = (\theta + \pi/2)$. Using quaternion operations, $\boldsymbol{\omega}_i^\lambda$ is then rotated around $\boldsymbol{\omega}_i$ by $\pi/2$ to derive $\boldsymbol{\omega}_i^\mu$. Then, we finally obtain the color with $c = c_d + c_s$ used in volume rendering equation, where $c_d$ is obtained by SHs from 3DGS (Kerbl et al., 2023).

**NeRSemble Real Dataset Division** We conduct experiments on video recordings of 9 subjects from the NeRSemble (Kirschstein et al., 2023) dataset as described on Tab. 4. As we focus on deformation, we select extreme expression set for self-reenactment to emphasize the difference. Therefore, among the emotion (EMO) and expression (EXP) sequences, we hold out EMO-1 for self-reenactment evaluation and use the rest nine for training as shown in Tab. 5.

| Train Sequences | EMO-2 EMO-3 EMO-4 EXP-2 EMO-3 EMO-4 EMO-5 EMO-8 EMO-9 |
|---|---|
| Test Sequences | EMO-1 |

Table 5: The train and test held-out sequences.

| Method | | GA | Base | + Jacobian | +JBS | +SGs (= Ours) |
|---|---|---|---|---|---|---|
| Rendering Speed (FPS) | +PyTorch | 71.18 | 109.36 | 107.59 | 92.62 | 90.13 |
| | +RoMA | N/A | N/A | N/A | 97.29 | 94.72 |
| Training Time (hours) | | 1.65 | 1.68 | 1.73 | 1.98 | 2.11 |

Table 7: Comparison of training and rendering (test) computational costs.

## A.4 ADDITIONAL EXPERIEMENT

**Comparsion with Gaussian Head Avatars (GHA) (Xu et al., 2024)** GHA serves as a strong baseline in rendering quality, leveraging an additional super-resolution model in screen space. As shown in Tab. 6, GHA outperforms our method in terms of PSNR. However, in other metrics such as SSIM and LPIPS, GHA falls behind.

Fig. 8 reveals potential reasons for this discrepancy. Notably, artifacts such as over-saturation are visible, which we attribute to GHA's screen-space refinement. This refinement struggles when rendering extreme poses that fall outside the distribution expected by the super-resolution model. Additionally, GHA faces challenges in reconstructing high-fidelity geometry and handling extreme expressions, both of which are key strengths of our approach. We would like to emphasize

Table 6: Quantitative comparsion with GHA (Xu et al., 2024)

| | PSNR↑ | SSIM↑ | LPIPS↓ | NCS ↑ |
|---|---|---|---|---|
| GHA | **27.25** | 0.909 | 0.153 | 0.505 |
| Ours | 26.20 | **0.932** | **0.052** | **0.837** |

the importance of capturing the specular highlights of the eyes, particularly the cornea. GHA renders the pupils with a matte appearance, neglecting the high-frequency specular reflections in the corneal region. We argue that these specular details are critical for enhancing the realism of the eyes, which play a pivotal role in creating immersive and lifelike head avatars.

## A.5 ADDITIONAL ABLATION STUDY

**Adaptive Density Control (ADC) Amplification on Teeth.** We posted qualitatives with the occlusion amplification on the teeth part in Fig. 9. This technique is very simple yet efficient with boosting Gaussian proliferation to overlooked parts in terms of optimization.

**Training and Rendering Time Complexity Measurement** Tab. 7 summarizes the rendering speed and training time for our method and GaussianAvatars (Kerbl et al., 2023). The Base configuration refers to replacing the 3D Gaussian Splatting rasterizers in GaussianAvatars with their 2D counterparts. Our method incurs only a 17% drop in rendering speed and an additional 25 minutes of training time, while still achieving $3\times$ real-time rendering speeds (generally over 30 FPS) and maintaining efficient training. A detailed analysis attributes the minimal training and testing overhead to our GPU-level CUDA kernel implementation for Jacobian computations.

For rendering, the primary factor behind the speed reduction is the Jacobian Blend Skinning (JBS), where the overhead mainly arises from the Polar Decomposition step. Our current implementation utilizes PyTorch's SVD, which relies on the cuSOLVER backend. To further investigate this bottleneck, we conducted additional experiments using RoMA (Brégier, 2021)'s specialized Procrustes routine, which is designed to efficiently compute the $3 \times 3$ unitary matrix $\mathbf{U}$ of the Jacobian $\mathbf{J}$. Notably, replacing `torch.svd` with `roma.special-procrustes` yielded a performance gain of approximately 4–5 FPS.

Although this demonstrates the potential of alternative approaches, there is still room for further improvement. Higham's routine (Higham & Noferini, 2016), specifically tailored for $3 \times 3$ matrices, offers a promising direction to address this overhead and is well-suited for CUDA-based implementations.

## A.6 LIMITATION AND SEMINAL FUTURE WORK

**Bounded Representation from 3DMFM.** We utilized a 3DMFM model, FLAME (Li et al., 2017) to enable our model to be equipped cross-reenactment and rigged compactly. However, the 3DMFMs' expression space has only PCA-based coarse rigging-ability such that cannot easily describe the dynamics of facial muscles such as wrinkles and extreme pose or expression (Fig. 10).

To end this, we believe that the image-based latent expression representations (Burkov et al., 2020; Wang et al., 2022; Drobyshev et al., 2022; 2024) can alleviate this issue some extent. However, naively applying the latent expression to dynamic head avatars is non-trivial. Because only designing dynamics of the geometrical deformation is hard to handle ambient-occlusion (such as shadow of neck and jaw from head and lips, each) without intricate physical-based rendering (PBR). Therefore, we guess that the future work should be conducted as learning the deformation dynamics with those latent expression spaces, also simultaneously covering the deformation-aware color changes.

Second, the inherent limitations of 3DMFM's representation, particularly the absence of hair strands and the tongue, can degrade the render quality. Since 3DMFM is fundamentally a bald model for human heads, it lacks the capability to represent the high-frequency details of human hair (Fig. 11). We believe that integrating our model with parametric strand models (Sklyarova et al., 2023; Zakharov et al., 2024; Rosu et al., 2022) could help alleviate these issues.

**Computational Efficiency for Calculating Polar Decomposition with Numerical Views.** Our Jacobian Blend Skinning (JBS) algorithm is based on polar decomposition, which is indirectly computed using byproducts of singular value decomposition (SVD). Specifically, $\mathbf{J} = \mathbf{W}\boldsymbol{\Sigma}\mathbf{V}^T$, where $\mathbf{W}$ and $\mathbf{V}$ are orthogonal matrices, and $\boldsymbol{\Sigma}$ is a diagonal matrix with singular values. This can be rewritten as $\mathbf{J} = (\mathbf{W}\mathbf{V}^T)(\mathbf{V}\boldsymbol{\Sigma}\mathbf{V}^T) = \mathbf{U}\mathbf{P}$, where $\mathbf{U}$ is the rotation matrix, and $\mathbf{P}$ is the positive semidefinite symmetric matrix.

However, as reported in RoMa (Brégier, 2021), PyTorch (Paszke et al., 2019)'s SVD routine scales linearly in time as the batch size increases. In the Gaussian Splatting (Kerbl et al., 2023; Huang et al., 2024) regime, over 10K operations for Gaussians are performed simultaneously to render an image, which requires the SVD routine to be executed in parallel. The polar decomposition process involves finding the nearest rotation matrix given a matrix. We believe that adopting RoMa (Brégier, 2021)'s Procrustes process or Higham's (Higham & Noferini, 2016) routine, which is specifically designed to solve problems for $3 \times 3$ matrices, could significantly improve efficiency.

**Extending SurFhead to Relighting Task.** Normal is one of the key intrinsic material properties. We demonstrate the possibility of using the normals generated by **SurFhead** for avatar relighting tasks in Fig. 13, utilizing GaussianShader (Jiang et al., 2024b). The relit results exhibit high plausibility, with high-fidelity reflections of the environment maps. We present objects reconstructed using our method, including relighting scenarios under various lighting conditions, featuring both warm and cool tones, as well as indoor and outdoor environments. Our renderings, across three diverse sets, convincingly demonstrate that the relit scenes maintain realism, exemplifying our method's proficiency in relighting applications.

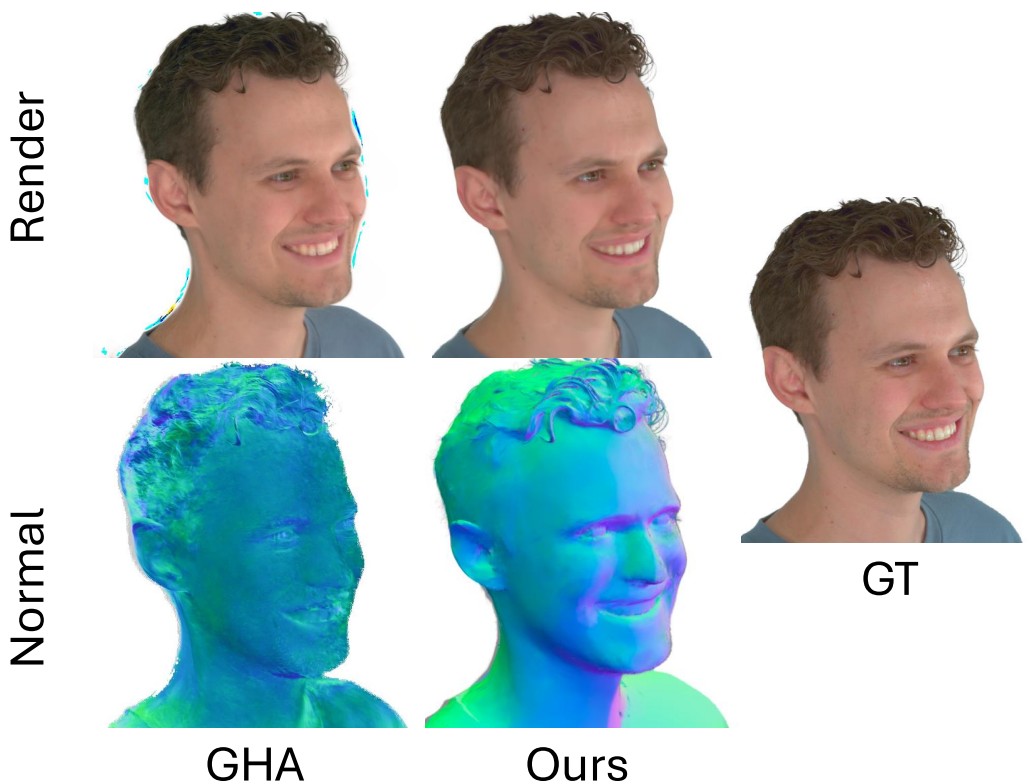

Figure 8: Qualitative Comparison with GHA (Xu et al., 2024)

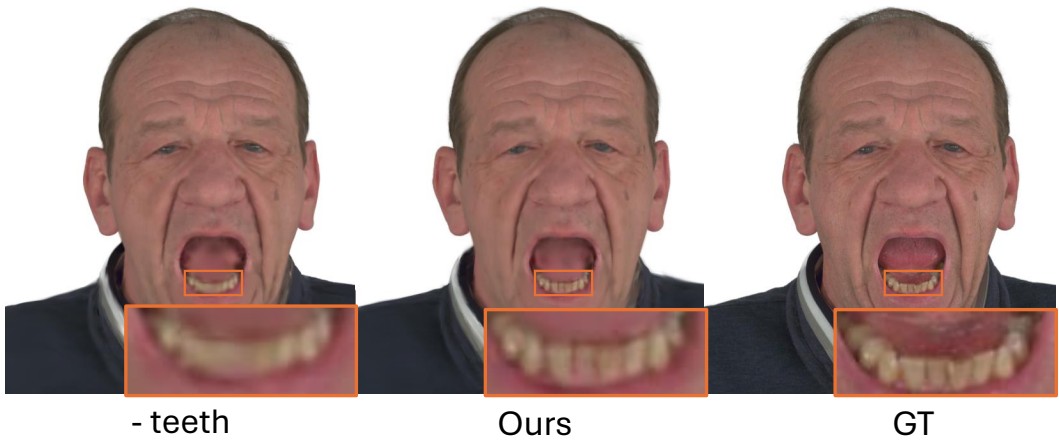

Figure 9: Ablations for ADC amplification on teeth.

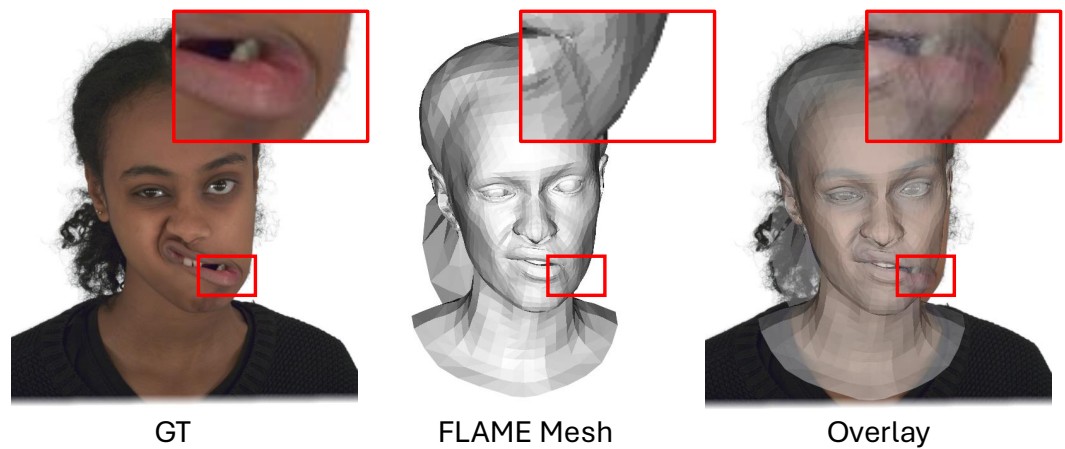

Figure 10: Limitation#1. Example for bounded representation of expression space from 3DMFM.

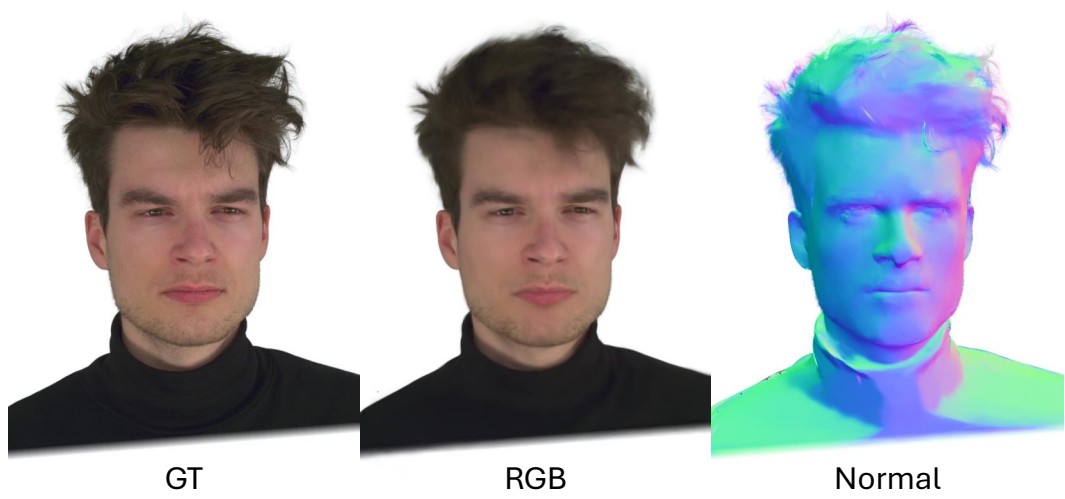

Figure 11: Limitation #2: The lack of hair modeling in 3DMFM can result in blurry hair appearance.

## A.7 CONCEPTUAL COMPARISON WITH PREVIOUS METHOD

Previous methods, (Qian et al., 2024; Shao et al., 2024), utilize isotropic scaling for deforming mesh-bound Gaussians, with scalar scaling parameter $s_p$. GaussianAvatars computed $s_p$ using the mean length of one of the edges and its perpendicular, while SplattingAvatars computed $s_p$ based on the ratio of canonical and deformed triangle areas. However, we find that these approaches can result in undesirable Gaussian deformations when the 3DMFM meshes undergo stretch or shear in specific directions. When the deformed and canonical triangle areas or the sum of an edge's length and its perpendicular are the same but the shape is differ, the Gaussians become misaligned and fail to cover the necessary regions accurately. In contrast, our method uses affine deformation, allowing Gaussians to more precisely reflect the deformation of the triangles and cover the regions as intended.

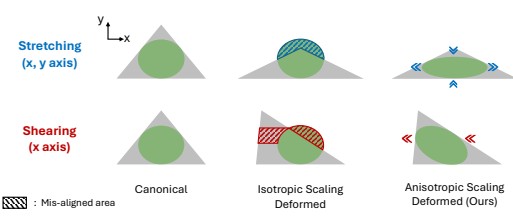

Figure 12: Conceptual illustration of stretching and shearing: **(Left)** rigid and isotropic scaling and **(Right)** affine deformation (ours)

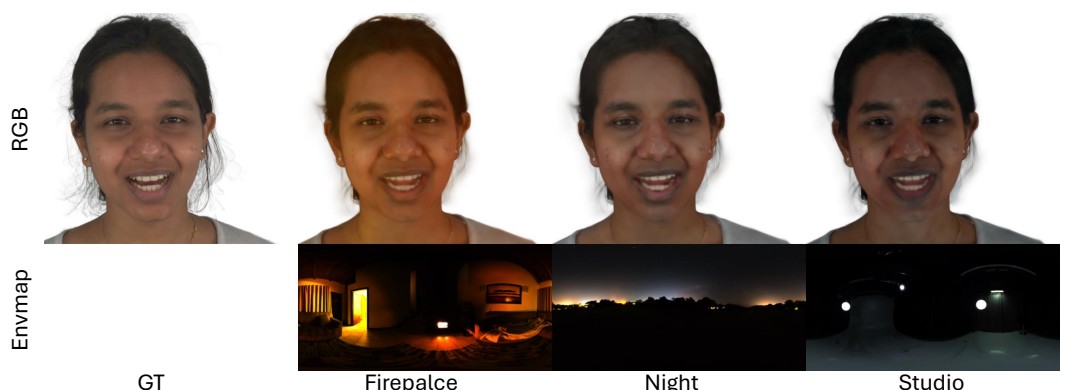

Figure 13: Results for relighting with GaussianShader (Jiang et al., 2024b).

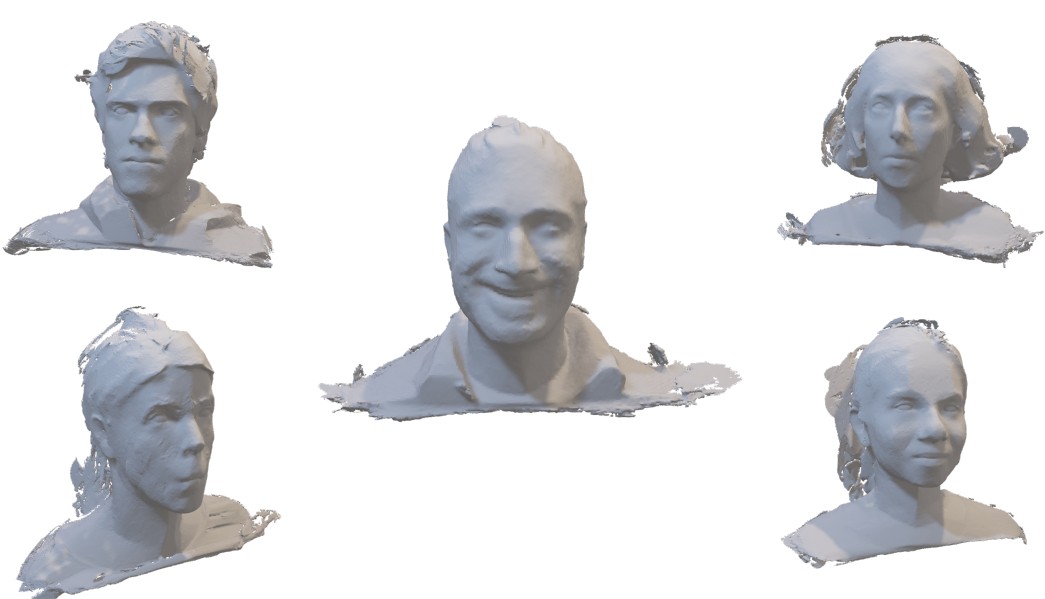

Figure 14: **SurFhead** can reconstruct high-fidelity meshes from Truncated Signed Distance Fucntion (TSDF) alongside diverse pose and expression.

