# OpenReview forum: "SurFhead: Affine Rig Blending for Geometrically Accurate 2D Gaussian Surfel Head Avatars"
_ICLR.cc/2025/Conference — ICLR 2025 Poster_

### Official Review · Reviewer_Cr49 · 2024-10-31

**Soundness:** 3
**Presentation:** 3
**Contribution:** 3
**Rating:** 6
**Confidence:** 3

**Summary:**

The paper proposes a method that handles stretch and shear transforms essential for detailed deformations of geometry utilizing intricate deformations driven by the affine Jacobian gradient instead of similarity transformation and corresponding normal adjustments.

**Strengths:**

1) Jacobian Blend Skinning (JBS): The use of Jacobian Blend Skinning (JBS) enables natural interpolation of affine transformations across adjacent deformations, effectively reducing discontinuities in transitions.

2) Cornea Opacity Constraint: To address the specular highlights in the eyeball region, the method constrains the corneal regions to remain opaque by regularizing the opacity of the respective Gaussians.

**Weaknesses:**

1) Detail Representation: In Figure 5 (bottom row), there seems to be a lack of finer details, such as wrinkles. Adding more visual comparisons or details on addressing such high-frequency features could strengthen the analysis.

2) Rendering Speed and FPS: Given that methods like 3DGS/2DGS achieve real-time rendering, the speed of deformable-driven methods may be a limitation for applications requiring real-time animation. Could you report the FPS compared to other methods to clarify performance in time-sensitive scenarios?

**Questions:**

1) Related Works: I think some related works of static reconstruction could be further discussed, such as H3DS [1], deformable model-driven approaches [2], and Implicit Neural Deformation methods [3]. These methods leverage 3D points from SfM, multi-scans, or multi-view data from various identities to enhance reconstruction under sparse-view conditions, although they primarily focus on static human face or head geometry reconstruction.

Ref:

[1] Ramon, Eduard, et al. "H3d-net: Few-shot high-fidelity 3d head reconstruction." Proceedings of the IEEE/CVF International Conference on Computer Vision. 2021.

[2] Xu, Baixin, et al. "Deformable model-driven neural rendering for high-fidelity 3D reconstruction of human heads under low-view settings." Proceedings of the IEEE/CVF International Conference on Computer Vision. 2023.

[3] Li, Moran, et al. "Implicit Neural Deformation for Sparse‐View Face Reconstruction." Computer Graphics Forum. Vol. 41. No. 7. 2022.

**Details Of Ethics Concerns:**

The authors have thoroughly discussed the potential ethics impact of detailed head avatars.

---

> ### Author Response · Authors · 2024-11-20
>
> We would like to thank the reviewer for acknowledging our work's effectiveness such as Jacobian Blend Skinning and Eyeball regularizations. Following section will be the responses of the reviewer's concerns, fine-detailed reconstruction and reporting time complexity.
>
> >
> > ## Fine-detailed Reconstruction Quality
> >
>
> Since the most bottom row of Fig. 5 shows cross-reenactment results which is not easy to validate the subject-centric deformations such as wrinkles. Therefore, to validate for solid manner, we show additional results which shows wrinkle-related geometric detail in [this link](https://surfhead2025.github.io/static/rebuttal/Cr49_264_wrinkles.png). As can be seen in the image file, the addition of Jacobian and Jacobian Blend Skinning shows increase of quality of wrinkles. Although this achievement, we regard there has a room for improving this fine-detailed parts. As we mentioned in the Section A.5 in Appendix, we suggested the future work related to this part. Notably, wrinkles and facial shadows are inherently influenced by ambient occlusion,
> which highlights the importance of dynamic appearance modeling in achieving realistic representations. Incorporating dynamic appearance features related pose and expression conditioned such as color and opacity, is essential. For related work, please refer to GHA **[1]**, HeadGAS **[2]**, and NPGA **[3]**. This could be also viable option for realistic rendering and geometry, but we intentionally excluded network-inferred color for potential time-complexity overheads. Exploring ways to effectively represent ambient occlusion while reducing such time-complexity overhead presents a highly interesting direction for future research. Nevertheless, we effectively captured deformations beyond 3DMM using JBS, which we would like to highlight as one of our achievement.
>
> >
> > ## Missing Related Work
> >
> Thanks for pointing out these related works and for insightful suggestions. We agree that discussing these methods would enrich the context of our work. We have added the literature mentioned by the reviewer to the revised version under the section `Neural Surface Reconstruction` in Related Work A.1 of the Appendix. Please refer the revised version of our paper.
>
> >
> > ## Training and Rendering Time Complexity Measurement
> >
> Please kindly refer to the answer ` Training and Rendering Time Complexity Measurement.` from the general response.
>
> We sincerely thank the reviewer once again for this valuable suggestion; the provided analysis are already incorporated into the current revised manuscript and appendix.
>
> We would greatly appreciate any further comments or discussions on potential improvements to enhance our manuscript.
>
> > *Reference*
>
> **[1]** Xu, Yuelang, et al. "Gaussian head avatar: Ultra high-fidelity head avatar via dynamic gaussians." Proceedings of the IEEE/CVF Conference on Computer Vision and Pattern Recognition. 2024.
>
> **[2]** Dhamo, Helisa, et al. "Headgas: Real-time animatable head avatars via 3d gaussian splatting." European Conference on Computer Vision. Springer, Cham, 2025.
>
> **[3]** Giebenhain, Simon, et al. "NPGA: Neural Parametric Gaussian Avatars." arXiv preprint arXiv:2405.19331 (2024).

---

> ### Author Response · Authors · 2024-11-23
> **We also thank you for your valuable feedbacks.**
>
> Thank you for the *R Cr49*'s positive feedback on the additional visualization results and for recognizing the key role of JBS in dynamic head reconstruction.
> We are pleased to hear that our response has addressed your concerns.
> Based on your comments, it seems that our paper now aligns with your expectations.
>
>
> If there are any remaining points of discussion or clarifications that could further strengthen your assessment of our work, we would be more than happy to address them.
> Any questions and discussions are welcome.
>
>
> Your constructive feedback has been invaluable, and we hope the revisions merit a higher rating reflecting our improvements!

---

> > ### Author Response · Authors · 2024-11-27
> >
> > We would like to express our gratitude for taking the time to thoroughly review our rebuttal session. We understand that the rebuttal session has been extended by one week, and we are fully committed to addressing any additional discussions or concerns you may have. Your feedback is invaluable to us as we strive to improve our manuscript and contribute to the academic community.
> >
> > Please do not hesitate to reach out if you have any further questions or suggestions. We are eager to engage in constructive dialogue and make any necessary revisions to ensure our work meets the high standards.
> >
> > Thank you once again for your valuable input.

---

### Official Review · Reviewer_RtVj · 2024-11-02

**Soundness:** 3
**Presentation:** 3
**Contribution:** 3
**Rating:** 6
**Confidence:** 2

**Summary:**

This paper contributes a novel representation for geometrically accurate head avatars within the 3D Gaussian Splatting framework, a method for natural interpolation of affine transformations across adjacent deformations, and enhancements to the realism of corneal representations in head avatars. These contributions advance the state of the art in personalized head avatar construction and have the potential to improve various applications in computer graphics, virtual reality, and beyond.  The key contributions include:

* Introduction of SurFhead Model: The paper introduces SurFhead, the first geometrically accurate head avatar model within the Gaussian Splatting framework. This model is designed to capture the deformation of head geometry using intricate affine rigging that combines Gaussians and their normals solely from RGB videos.

* Jacobian Blend Skinning Algorithm: To address the issue of discontinuities between adjacent triangles in head deformations, the paper proposes the Jacobian Blend Skinning (JBS) algorithm. This algorithm blends adjacent transformations while avoiding geometric distortions by linearizing the non-linear matrix interpolation space, leveraging classical matrix animation techniques and geometrically smooth polar decomposition.

* Enhancement of Corneal Convexity and Specularity: The paper addresses the hollow illusion in the cornea by regularizing corneal convexity and enhancing specularity using computationally efficient Anisotropic Spherical Gaussians (ASGs). This improvement ensures a more realistic representation of the cornea in the head avatar.

**Strengths:**

The paper introduces SurFhead, a novel model within the Gaussian Splatting framework that captures geometrically accurate head deformations. This representation utilizes intricate affine rigging combined with Gaussians and their normals, solely based on RGB videos, which is a significant advancement in achieving realistic and detailed head avatars.  The proposed Jacobian Blend Skinning (JBS) Algorithm is technically sound.  The paper tackles the problem of the hollow illusion in the cornea, where a concave surface appears convex due to the prioritization of photometric losses during training.

The methods presented in the paper are demonstrated to achieve superior results across a variety of subjects, including real and synthetic data. They excel in challenging scenarios such as sharp reflections on convex eyeballs, fine geometric details, and exaggerated deformations, showcasing the robustness and effectiveness of the proposed approach.

**Weaknesses:**

The method is built upon the foundation of 2D Gaussian Splatting, and I believe that the proposed method's success in recovering a superior surface geometry owes much to this solid groundwork. Indeed, the authors introduced several improvements on this basis, such as intricate affine rigging, but I consider these innovations to be more incremental improvements rather than groundbreaking advancements.

The geometric accuracy of the experimental results appears to exhibit significant variation: some achieve hair-level geometric detail, while others fail to recover the structure of the hair. Consequently, whether this variation stems from instability in the algorithm or differences in the data quality of various training datasets arises. I hope the author can provide more analysis and discussion on this issue.

**Questions:**

As elaborated in the "weaknesses", the geometric accuracy of the experimental results exhibits considerable variation,  and we hope that the authors can further enhance their analysis and discussion on this crucial point.

In terms of experimental design, the NeRSemble dataset, while containing accurate 3D mesh models, lacks sufficient detail, and obtaining such precise models can often be challenging in practical applications.     In this regard, we eagerly inquire whether the proposed method heavily relies on such relatively accurate 3D mesh models, and how it would perform in their absence.     To validate this, the authors could consider using videos they have shot or sourced from the internet as input, employing monocular facial 3D reconstruction algorithms (such as DECA) to obtain mesh sequences, or directly bypassing the use of 3D mesh models altogether.     We are anticipating and curious about the results of such experimental setups.    We encourage the authors to actively explore and propose potential strategies for adapting their proposed approach to scenarios where detailed 3D mesh models are unavailable.

---

> ### Author Response · Authors · 2024-11-20
>
> We would like to thank the reviewer for considering our work as exceptionally novel and thoroughly evaluated and ablated, as well as for the insightful comments and questions regarding the proposed method and its performance. In the following, we address the remaining discussion points related to our innovation, the potential effect of hair-level geometric quality of proposed framework, and utilizing off-the-shelf head tracking algorithm with monocular videos.
>
> >
> > ## Innovation of SurFHead
> >
>
> We would like to draw attention to Tab. 3, which highlights the significant improvements over the vanilla 2DGS **[1]** (eyeball regularization added). Specifically, the increase in PSNR from 22.35 to 23.09 is substantial, given that PSNR is measured on a logarithmic scale. Not only PSNR, SSIM, LPIPS, and NCS shows significant improvement compared with vanilla 2DGS. Beyond this improvement, our method introduces valuable mathematical properties, including Jacobian gradients, normal deformation, and Jacobian Blend Skinning (JBS).
>
> Furthermore, we believe these properties lay a strong foundation for advancing dynamic Gaussian models, extending their applications beyond head avatars to encompass full-body, hands and 4D Gaussian representations. As noted by reviewer *D2NV*, we believe that these advancements can serve as meaningful cornerstones for future research.
>
> >
> > ## Robustness of Geometric Quality of Hair Strands
> >
>
> Hair reconstruction and simulation are among the most challenging aspects of modeling the human head. We believe the variability in hair reconstruction results is primarily due to the nature of the data. When the head is simultaneously talking and swaying, the dynamic motion of hair strands introduces instability during training, often resulting in an averaged representation of the moving strands. This issue arises from the inherent elasticity and non-rigidity of hair strands, as demonstrated in [the attached video](https://drive.google.com/file/d/18mIeD7UoLvj9GWAFKm5_FGi0fxRNvCCC/view?usp=drive_link). Even after the subject's head stops moving, several strands continue to oscillate due to their elastic properties. This phenomenon is illustrated in Fig. 10 of the Appendix. To further support our hypothesis, we included the results of GaussianAvatars with Ours in [this link](https://surfhead2025.github.io/static/rebuttal/RtVj_Hair.png). This result demonstrates that using 3D Gaussians instead of 2D surfels faces the same issue.
>
> >
> > ## Train with Coarse Mesh (monocular datasets)
> >
> To evaluate the extent to which our method relies on preprocessing, as requested by the reviewers, we trained on a monocular dataset using a low-cost keypoint-based tracking approach. The qualitative results, which can be viewed at [this link](https://surfhead2025.github.io/static/rebuttal/RtVj_monocular_malte_1.png), demonstrate that while monocular datasets inherently face challenges in reconstructing normals for occluded areas (such as below the jaw or side profile) compared to multiview datasets, this limitation is not specific to our method but rather a consequence of the narrow field of view in monocular datasets. Nonetheless, SurFhead achieves superior performance compared to GaussianAvatars in capturing fine details such as wrinkles and provides significantly better normal estimations.
>
>
> We sincerely thank the reviewer once again for this valuable suggestion; the provided analysis are already incorporated into the current revised manuscript and appendix.
>
> We would greatly appreciate any further comments or discussions on potential improvements to enhance our manuscript.
>
> >*Reference*
>
> **[1]** Huang, Binbin, et al. "2d gaussian splatting for geometrically accurate radiance fields." ACM SIGGRAPH 2024 Conference Papers. 2024.

---

> > ### Author Response · Authors · 2024-11-25
> > **Follow-Up on Updated Manuscript and Responses**
> >
> > Dear Reviewer RtVj,
> >
> > We wanted to kindly follow up to inquire if you’ve had the chance to review our updated manuscript and the responses addressing your discussion points. Your feedback is extremely valuable to us, and we would greatly appreciate any additional questions or comments you may have.
> >
> > Thank you for your time and consideration.
> >
> > Best regards,
> > The Authors

---

> > ### Comment · Reviewer_RtVj · 2024-11-26
> >
> > Thank the authors for the response. The authors demonstrated the performance of SurFhead trained on monocular datasets with coarse meshes, dismissing my concerns about the method's robustness. I still insist on the original evaluation of the proposed method's novelty. Overall, I maintain the rating of "positive above the acceptance threshold".

---

> > > ### Author Response · Authors · 2024-11-27
> > >
> > > We would like to express our gratitude for taking the time to thoroughly review our rebuttal session. We understand that the rebuttal session has been extended by one week, and we are fully committed to addressing any additional discussions or concerns you may have. Your feedback is invaluable to us as we strive to improve our manuscript and contribute to the academic community.
> > >
> > > Please do not hesitate to reach out if you have any further questions or suggestions. We are eager to engage in constructive dialogue and make any necessary revisions to ensure our work meets the high standards of your expectations.
> > >
> > > Thank you once again for your valuable input.

---

### Official Review · Reviewer_u9qy · 2024-11-02

**Soundness:** 3
**Presentation:** 3
**Contribution:** 3
**Rating:** 8
**Confidence:** 4

**Summary:**

This paper aims at better geometry estimation in head modeling. It replaces the 3D Gaussian splatting with 2D Gaussian splatting to better model the surface. Moreover, it addresses three issues in existing works with three novel components: 1) To compensate for the incorrect deformation of 2D Gaussians due to the triangle's shear and stretch, it proposes the Jacobian deformation; 2) To mitigate the discontinuities in adjacent triangles, it improves linear blend skinning with Jacobian blend skinning; 3) To resolve hollow illusion in eyeballs, it replaces Spherical Harmonics with Anisotropic Spherical Gaussians. It demonstrates that it outperforms the state-of-the-art regarding normal similarity and remains comparable in rendering quality.

**Strengths:**

* It motivates each contribution properly.
* The use of Jacobian deformation and Jacobian blend skinning in the context of head modeling looks novel to me.
* The qualitative results and normal similarity evaluation demonstrate better geometry compared to baselines.
* The effectiveness of each component is well studied.

**Weaknesses:**

* It is unclear how the deformation gradient $J$ in Sec 2.2 and the blended Jacobian $J_b$ in Sec 2.3 connect. In Line 258 to 261, it mentions that it replaces the original deformation in GaussianAvatar with a new deformation $J_b$, and $J$ appears only as a parameter of JBS in Eq. 2. What is the relationship between $J$ and $U_i, P_i$ in Eq. 2? Which transformation, $J$ or $J_b$, is used in the final method?
* The paper measures the normal similarity between ground-truth normals and rendered normals from 2D Gaussians. However, since this work claims to achieve a better geometry, evaluating metrics that apply to meshes, such as Chamfer distance or normals rendered from the mesh is more informative when judging the geometry. Although previous methods use normals rendered from 2D Gaussians as proof of geometry, the link between it and the mesh quality still looks vague to me.
* As the rendering quality is only on par with state-of-the-art, e.g., GaussianAvatar, the comparison of training and rendering speed is missing.

**Questions:**

* It is unclear to me how Jacobian blend skinning improves upon linear blending skinning. It seems to me that $J_b$ introduces spatial smoothness to Gaussians' deformation matrices. Is the vertices of the mesh still transformed by linear blending skinning before the Jacobian blend skinning is applied? Also, could the author clarify Fig. 2(b)? What are the meanings of the green and yellow lines and the weights? How do they connect to the deformation of the triangle meshes?
* The paper mainly focuses on improving the geometry. Since 2D Gaussian splatting is known to produce better geometry than 3D Gaussians, how much does 2D Gaussians help improve the geometry, compared to the components proposed in the paper?

---

> ### Author Response · Authors · 2024-11-20
>
> We thank the reviewer for acknowledging our work's contribution, superiority and novelty, and that its effectiveness is well studied. In the following, we address the remaining discussion points:
>
> >
> > ## Clarifying again the role of $\mathbf{J}_{b}$
> >
>
> For the first, our final parameterization is usage of $\mathbf{J}\_{b}$, not $\mathbf{J}$, as mentioned in Eq. (3) and (4).
> To give more understanding, the GaussianAvatars' square root of covariance $\Sigma^{1/2} $is defined as $RS = s_{p}R_{p}R_{c}S_{c}$ and position $\mu = s_{p}R_{p}\mu_{p}$, as mentioned in Eq. (1). To handle the shear and stretch deformations, we introduced Jacobian Gradient $\mathbf{J}$ which replaces $s_{p}R_{p}$ term in square root of covariance and position. This formula is also could be found in 214th line. Finally, to tackle the potential discontinuity of local deformations, we replace the Jacobian Gradient $\mathbf{J}$ to blended Jacobian $\mathbf{J}\_{b}$, again. This is clarified in Eq. (3). Specifically, the $\mathbf{J}$ is broken down in to unitary matrix $\mathbf{U}$ and positive semi-definite matrix $\mathbf{P}$. The $\mathbf{J}\_{b}$ is derived from blending with adjacent triangles $\mathbf{U}$s and $\mathbf{P}$s such as Eq. (2).
> To summarize, there are two transitions for transformations: $s_{p}R_{p}$ $\rightarrow$ $\mathbf{J}$ $\rightarrow$ $\mathbf{J}_{b}$. Note that this deformation is only applied for Gaussians, not mesh triangles which provide the base deformation such as $\mathbf{J}$.
>
> I hope that this explanation helped the reviewer's comprehension. Also, we augmented these flow of transition to current revised version in Section 2.3. Are there any other questions, please leave comments to let us know. We will reply as soon as possible.
>
> >
> > ## Reliable Metric to Evaluate Geometry Quality.
> >
> Thanks for your insightful comment. As suggested, we measured the Chamfer distance on the Facetalk dataset `id-8`, which includes ground truth meshes. Since it is a monocular dataset, we focused on the frontal region and randomly sampled 30 cameras near the frontal view (both azimuth and elevation sampled from $U\sim(-0.8, 0.8)$ in radian scale) to extract meshes using a Truncated Sign Distance Field (TSDF) method.
> We compared our approach with both FLARE and GaussianAvatars. FLARE was chosen for comparison because it has the second-highest Normal Cosine Similarity (NCS) after our method as shown in Tab. 1. For FLARE, which is a mesh-based method, we directly used its optimized mesh for comparison. For GaussianAvatars, we extracted the mesh using the same TSDF-based method as our approach. As shown in the table below, the Chamfer Distance of the meshes generated by our proposed method outperformed both FLARE and GaussianAvatars. This indicates that our approach achieves an improvement also in mesh quality compared to these methods.
>
> |Method|Chamfer Distance|
> |--------|-----|
> |FLARE|0.1590|
> |GA|0.4366|
> |Ours|**0.1571**|
>
> >
> > ## Additional Qualitative Results in Extreme Scenarios
> >
>
> As demonstrated in Fig. 5, we have showcased the qualitative rendering and geometry quality of our method under extreme pose scenarios. To quantitatively evaluate the robustness of our approach in such scenarios, we selected the top 10 largest jaw poses and expressions per subject, totaling 20 images per subject, and conducted evaluations across 9 subjects.
>
> As shown in the table below, our method performs on par with GaussianAvatars in terms of both rendering and geometry quality, demonstrating its effectiveness under challenging conditions.
>
> |Method|PSNR|SSIM|LPIPS|NCS|
> |----|----|-----|-----|-----|
> |GaussianAvatars|29.60|0.912|0.088|0.702|
> |Ours|**29.79**|**0.925**|**0.082**|**0.883**|
>
> >
> > ## Effect of Changing 3D Gaussians to 2D Gaussians
> >
> Naively replacing 3D Gaussians with 2D Gaussians improves geometric quality but does not enhance rendering quality, as shown in the table below. This trend has also been observed in 2DGS **[1]**. However, our method surpasses these limitations, demonstrating superior rendering and geometry quality. This is achieved through the proposed Jacobian-based normal deformation, Jacobian Blend Skinning (JBS), and the incorporation of eyeball constraints.
> Please  refer the Table 3. in the current revised version of the manuscript.
>
> |Method|PSNR|SSIM|LPIPS|NCS|
> |----|----|-----|-----|-----|
> |GaussianAvatars|**22.49**|**0.920**|**0.089**|0.727|
> |GaussianAvatars+2DGS |22.32|0.907|0.093|**0.803**|
>
>
> > *Reference*
>
> **[1]** Huang, Binbin, et al. "2d gaussian splatting for geometrically accurate radiance fields." ACM SIGGRAPH 2024 Conference Papers. 2024.

---

> > ### Author Response · Authors · 2024-11-20
> >
> > >
> > > ## Clarifying Fig. 2 (b).
> > >
> >
> > Figure 2-(b) illustrates the flaw of direct element-wise interpolation (in matrix space), where the star shape collapses into a line or a point. Specifically, this can be observed at the label (0.5,0.5) in the figure. The yellow and green lines are included for improved visibility, as the 5-fold symmetry of the star shape makes it difficult to discern which vertex is being rotated. The above parenthesis label indicates the weight of each left and right-most matrix.
> >
> > The intuitive reason for this collapse during direct interpolation is that it is equivalent to connecting a straight line between the starting and ending matrices in Euclidean space. Consequently, the interpolated shape lies along the internally dividing point of that line, leading to the observed deformation. This phenomenon is analogous to the *candy-wrapper* effect commonly seen in Linear Blend Skinning (LBS), as discussed in Dual Quaternion Skinning **[1]**.
> >
> > In contrast, our proposed Jacobian Blend Skinning (JBS) separates the rotation and positive semi-definite components of the Jacobian, blending them within their respective spaces—Rodrigues for rotation and a linear space for the positive semi-definite part. This approach effectively and mathematically robustly avoids such artifacts, preserving the integrity of the interpolated shape.
> >
> > >
> > > ## Training and Rendering Time Complexity Measurement
> > >
> > For training and rendering speed, please kindly refer to the answer ` Training and Rendering Time Complexity Measurement.` from the general response.
> >
> > We also revised Figure 2-(b) in revised version of the manuscript to make it easier to understand.
> >
> > We sincerely thank the reviewer once again for this valuable suggestion; the provided analysis are already incorporated into the current revised manuscript and appendix.
> >
> > We would greatly appreciate any further comments or discussions on potential improvements to enhance our manuscript.
> >
> > > *Reference*
> >
> > **[1]** Kavan, Ladislav, et al. "Skinning with dual quaternions." Proceedings of the 2007 symposium on Interactive 3D graphics and games. 2007.

---

> > > ### Comment · Reviewer_u9qy · 2024-11-24
> > >
> > > Thanks for the detailed responses. The authors addressed most of my concerns. The details are clear to me after revision. Overall, the paper provides reasonable solutions to issues in existing methods and shows promising results. After going through all the reviews, I would like to raise the score to 8.

---

> > > > ### Author Response · Authors · 2024-11-27
> > > >
> > > > We would like to express our gratitude for taking the time to thoroughly review our rebuttal session. We understand that the rebuttal session has been extended by one week, and we are fully committed to addressing any additional discussions or concerns you may have. Your feedback is invaluable to us as we strive to improve our manuscript and contribute to the academic community.
> > > >
> > > > Please do not hesitate to reach out if you have any further questions or suggestions. We are eager to engage in constructive dialogue and make any necessary revisions to ensure our work meets the high standards.
> > > >
> > > > Thank you once again for your valuable input.

---

### Official Review · Reviewer_D2NV · 2024-11-08

**Soundness:** 3
**Presentation:** 3
**Contribution:** 3
**Rating:** 6
**Confidence:** 4

**Summary:**

This paper presents a method for learning head avatars based on 2D Gaussian splatting. To make the Gaussian surfels better handle the stretch and shear deformation under extreme poses and facial expressions, this paper introduces affine transformation derived from Jacobian deformation gradient of the surface. Normal orientations are calculated accordingly. Moreover, the authors propose Jacobian blend skinning to interpolate these affine transformations to ensure surface smoothness. Results show that the proposed method is able to reconstruct drivable head avatars with high-quality geometry.

**Strengths:**

* This paper introduces better deformation modeling techniques for Gaussian surfels. Compared to existing works, the proposed method is more reasonable and can handle more extreme deformations such as stretching and shearing. The proposed technique could be useful in other related research topics beyond head avatar modeling.

* The proposed method is able to reconstruct fine geometric details, outperforming existing baselines by a large margin.

* The paper is overall well-written and easy to follow.

**Weaknesses:**

* Missing comparison against Gaussian Head Avatar (GHA) [Xu et al. 2023], which is a state-of-the-art head avatar method in terms of image synthesis quality. Although the authors have already compared with SplattingAvatar and GaussianAvatar, I think an additional comparison against GHA is also necessary because GHA demonstrates high-resolution image synthesis with the assistance of a super-resolution module.

* It would be better if the authors report the training time and rendering speed. One important advantage of Gaussian splatting is its efficiency. I wonder whether the proposed techniques (such as Jacobian blend skinning) hinders this advantage or not.

* It is not clear how the proposed method performs for subjects wearing eye-glasses. NeRSemble dataset contains cases with eye-glasses, but they are suspiciously skipped in the experiments.

**Questions:**

See [Weaknesses].

---

> ### Author Response · Authors · 2024-11-20
>
> We sincerely thank the reviewer for recognizing the effectiveness and versatility of our work beyond human head avatars, as well as for the insightful comments and experiment suggestions. Below, we address the feedback by contextualizing the SurFhead approach with the suggested related method, providing quantitative and qualitative comparisons with Gaussian Head Avatars (GHA) **[1]**, training and rendering speed evaluations, and additional experiments with eyeglasses-wearing subjects.
>
> >
> > ## **Comparison with GHA.**
> >
> We acknowledge that GHA serves as a strong baseline in rendering quality, leveraging an additional super-resolution model in screen space. To facilitate comparison, we provide additional quantitative and qualitative results in the table and images below. As shown in the table, GHA outperforms our method in terms of PSNR. However, in other metrics such as SSIM and LPIPS, GHA falls behind.
>
> Please be aware of [this link](https://surfhead2025.github.io/static/rebuttal/D2NV_GHA.png). The attached images reveal potential reasons for this discrepancy. Notably, artifacts such as over-saturation are visible, which we attribute to GHA's screen-space refinement. This refinement struggles when rendering extreme poses that fall outside the distribution expected by the super-resolution model. Additionally, GHA faces challenges in reconstructing high-fidelity geometry and handling extreme expressions, both of which are key strengths of our approach. We would like to emphasize the importance of capturing the specular highlights of the eyes, particularly the cornea. GHA renders the pupils with a matte appearance, neglecting the high-frequency specular reflections in the corneal region. We argue that these specular details are critical for enhancing the realism of the eyes, which play a pivotal role in creating immersive and lifelike head avatars.
>
> We will augment these results with additional subjects in Fig. 5 and Tab. 2 of the manuscript for the camera-ready version.
>
> |Method|PSNR|SSIM|LPIPS|NCS|
> |-----|---|-----|-----|-----|
> |GHA|**27.25**|0.909|0.153|0.505|
> |Ours|26.2|**0.932**|**0.052**|**0.837**|
>
> >
> > ## **Train with eyeglasses wearing identity.**
> >
> We noticed that the dataset provided by GaussianAvatar does not include data for subjects wearing glasses, so we had to preprocess new data accordingly. Additionally, it took some time to gain access to the NeRSemble dataset, and we are currently in the process of preprocessing it. We will share the results with you as soon as possible. Thank you for your understanding and patience!
>
> >
> > ## **Training and Rendering Time Complexity Measurement**
> >
>
> Please kindly refer to the answer ` Training and Rendering Time Complexity Measurement.` from the general response.
>
>
> We sincerely thank the reviewer once again for this valuable suggestion; the provided analysis are already incorporated into the current revised manuscript and appendix.
>
> We would greatly appreciate any further comments or discussions on potential improvements to enhance our manuscript.
>
> > *Reference*
>
> **[1]** Xu, Yuelang, et al. "Gaussian head avatar: Ultra high-fidelity head avatar via dynamic gaussians." Proceedings of the IEEE/CVF Conference on Computer Vision and Pattern Recognition. 2024.

---

> ### Author Response · Authors · 2024-11-23
> **(Added) Train with eyeglasses wearing identity**
>
> We appreciate your time and attention.
>
> We have successfully trained the eyeglasses-wearing subject (NeRSemble - 079). As demonstrated in the [video](https://surfhead2025.github.io/static/rebuttal/eyeglass_color_079.mp4) and [image](https://surfhead2025.github.io/static/rebuttal/eyeglass_079.png) showcasing the self-reenactment task, our method achieves high-quality geometry reconstruction and rendering. Even for a subject wearing eyeglasses, our approach faithfully reproduces both the convex eyeballs and the eyeglasses with high fidelity.
>
> While our method does not explicitly model eyeglasses, such as capturing the metallic frame's reflectivity or the specular effects of the lenses, these aspects are not the central focus of the dynamic head avatar task. Future work could explore integrating a parametric eyeglasses modeling framework, such as MEGANE **[1]**, to create a unified avatar system.
>
> >
> >*Reference*
> >
> **[1]** Li, Junxuan, et al. "Megane: Morphable eyeglass and avatar network." Proceedings of the IEEE/CVF Conference on Computer Vision and Pattern Recognition. 2023.

---

> ### Author Response · Authors · 2024-11-25
> **Follow-Up on Updated Manuscript and Responses**
>
> Dear Reviewer D2NV,
>
> We wanted to kindly follow up to inquire if you’ve had the chance to review our updated manuscript and the responses addressing your discussion points. Your feedback is extremely valuable to us, and we would greatly appreciate any additional questions or comments you may have.
>
> Thank you for your time and consideration.
>
> Best regards,
> The Authors

---

> ### Comment · Reviewer_D2NV · 2024-11-25
> **Reviewer Feedback**
>
> Thanks for the response. After reading the response and other reviews, I would like to maintain my rating.

---

> > ### Author Response · Authors · 2024-11-27
> >
> > We would like to express our gratitude for taking the time to thoroughly review our rebuttal session. We understand that the rebuttal session has been extended by one week, and we are fully committed to addressing any additional discussions or concerns you may have. Your feedback is invaluable to us as we strive to improve our manuscript and contribute to the academic community.
> >
> > Please do not hesitate to reach out if you have any further questions or suggestions. We are eager to engage in constructive dialogue and make any necessary revisions to ensure our work meets the high standards.
> >
> > Thank you once again for your valuable input.

---

### Author Response · Authors · 2024-11-20
**General Response**

We would like to thank all reviewers for their constructive feedback and insightful comments. We especially appreciate that reviewers consider our work novel and sound *(R u9qy, R RtVj)*, effectiveness proved *(R D2NV, R u9qy, R RtVj, R Cr49)*,  thoroughly evaluated *(R D2NV, R u9qy, R RtVj)*, and versatile *(R D2NV)*.

We have responded to the comments of each reviewer and subsequently revised both our manuscript and Appendix, as outlined below.

> ### To R D2NV
#### 1. We compared SurFhead with an additional baseline, GHA [1].
#### 2. We trained an additional subject wearing eyeglasses (to be conducted as soon as possible).

> ### To R u9qy
#### 1. We clarified the role of $\mathbf{J}_{b}$.
#### 2. We adopted another metric, Chamfer Distance, to evaluate geometric quality.
#### 3. We provided additional qualitative results in extreme scenarios.
#### 4. We added further explanation of Fig. 2(b).
#### 5. We analyzed the effect of changing 3D Gaussians to 2D Gaussians.

> ### To R RtVj
#### 1. We highlighted the innovation of SurFhead.
#### 2. We analyzed the robustness of the geometric quality of hair strands.
#### 3. We trained SurFhead on monocular datasets with coarse meshes.

> ### To R Cr49
#### 1. We analyzed the fine-detailed reconstruction quality.
#### 2. We addressed the missing related work.

We sincerely thank our reviewers and look forward to further discussions.


Below is the training time and test time comparison *(R D2NV, R u9qy, R Cr49)* to highlight the efficiency of our method.

>
> ## **Training and Rendering Time Complexity Measurement**
>

The table below summarizes the rendering speed and training time for our method and GaussianAvatars **[2]**. The *Base* configuration refers to replacing the 3D Gaussian Splatting rasterizers in GaussianAvatars with their 2D counterparts. Our method incurs only a **17\% drop in rendering speed** and an **additional 25 minutes of training time**, while still achieving **3$\times$ real-time rendering speeds** (generally over 30 FPS) and maintaining efficient training. A detailed analysis attributes the minimal training and testing overhead to our GPU-level CUDA kernel implementation for Jacobian computations.

For rendering, the primary factor behind the speed reduction is the *Jacobian Blend Skinning (JBS)*, where the overhead mainly arises from the Polar Decomposition step. As described in Section A.3 of the Appendix, our current implementation utilizes PyTorch's SVD, which relies on the cuSOLVER backend. To further investigate this bottleneck, we conducted additional experiments using RoMA **[3]**  's specialized Procrustes routine, which is designed to efficiently compute the $3\times3$ unitary matrix $\mathbf{U}$ of the Jacobian $\mathbf{J}$. Notably, replacing `torch.svd` with `roma.special_procrustes` **yielded a performance gain of approximately 4–5 FPS.**

Although this demonstrates the potential of alternative approaches, there is still room for further improvement. Higham's routine **[4]** , specifically tailored for $3 \times 3$ matrices, offers a promising direction to address this overhead and is well-suited for CUDA-based implementations.

| Method           | GA   | Base   | +Jacobian | +JBS  | +eyeballs (=Ours) |
|------------------|-------|--------|-----------|--------|-------------|
| **Rendering Speed (FPS)**                                                                 |
| `torch.svd`        | 71.18 | 109.36 | 107.59    | 92.62  | 90.13       |
|  `roma.special_procrustes`           | N/A   | N/A    | N/A       | 97.29  | 94.72       |
| **Training Time (hours)** | 1.65   | 1.68    | 1.73       | 1.98   | 2.11       |



> *Reference*

**[1]** Xu, Yuelang, et al. "Gaussian head avatar: Ultra high-fidelity head avatar via dynamic gaussians." Proceedings of the IEEE/CVF Conference on Computer Vision and Pattern Recognition. 2024.

**[2]** Qian, Shenhan, et al. "Gaussianavatars: Photorealistic head avatars with rigged 3d gaussians." Proceedings of the IEEE/CVF Conference on Computer Vision and Pattern Recognition. 2024.

**[3]** Brégier, Romain. "Deep regression on manifolds: a 3d rotation case study." 2021 International Conference on 3D Vision (3DV). IEEE, 2021.

**[4]** Higham, Nicholas J., and Vanni Noferini. "An algorithm to compute the polar decomposition of a 3× 3 matrix." Numerical Algorithms 73 (2016): 349-369.

---

### Meta-Review · Area_Chair_HJiQ · 2024-12-18

**Metareview:**

The paper introduces a new model within the Gaussian Splatting framework designed to capture realistic and detailed head deformations from RGB videos. The model enhances geometric accuracy and detail, particularly in complex scenarios like sharp reflections and exaggerated deformations, with good efficiency. The authors provide a substantial improvement over existing methods in terms of geometric detail reconstruction.

***Strengths:***
- The introduction of Jacobian deformation and blend skinning is novel and effectively enhances the model's ability to handle complex - deformations like stretching and shearing.
- The proposed method reconstructs fine geometric details more accurately than existing baselines.
- The paper is clearly written and easy to follow, aiding in its comprehension and potential replication.

***Weaknesses:***
The lack of results like comparison with GHA, glasses wearing cases, robust demonstrations etc.
Missing details on training time and rendering speeds.

Despite some concerns, the reviewers are generally positive, highlighting the paper’s innovative approach and its effectiveness in handling complex scenarios. The authors have also adequately addressed the issues with added experimental results.

After careful discussion and consideration, we are pleased to inform this paper is accepted. The paper is accepted based on its contributions to geometric detail reconstruction in head avatars, its novel methodological advancements, and the overall positive reception from reviewers.

**Additional Comments On Reviewer Discussion:**

Most of the questions and concerns are from lack of experimens and lack of details. The authors have addressed them well in the rebuttal. One reviewer raised score to 8. Overall evaluatin from all reviewers are positive for its effectiveness and maintaining efficiency.

---

### Decision · Program_Chairs · 2025-01-22

Accept (Poster)